# Global temperature modes shed light on the Holocene temperature conundrum

Jürgen Bader [1,2 ✉], Johann Jungclaus[1], Natalie Krivova [3], Stephan Lorenz[1], Amanda Maycock [4], Thomas Raddatz[1], Hauke Schmidt [1], Matthew Toohey [5,6], Chi-Ju Wu[3] & Martin Claussen [1,7]

Reconstructions of the global mean annual temperature evolution during the Holocene yield conflicting results. One temperature reconstruction shows global cooling during the late Holocene. The other reconstruction reveals global warming. Here we show that both a global warming mode and a cooling mode emerge when performing a spatio-temporal analysis of annual temperature variability during the Holocene using data from a transient climate model simulation. The warming mode is most pronounced in the tropics. The simulated cooling mode is determined by changes in the seasonal cycle of Arctic sea-ice that are forced by orbital variations and volcanic eruptions. The warming mode dominates in the mid-Holocene, whereas the cooling mode takes over in the late Holocene. The weighted sum of the two modes yields the simulated global temperature trend evolution. Our findings have strong implications for the interpretation of proxy data and the selection of proxy locations to compute global mean temperatures.

[1] Max-Planck-Institut für Meteorologie, Hamburg, Germany. [2] Uni Climate, Uni Research & the Bjerknes Centre for Climate Research, Bergen, Norway. [3] Max-Planck-Institut für Sonnensystemforschung, Göttingen, Germany. [4] School of Earth and Environment, University of Leeds, Leeds, UK. [5] GEOMAR Helmholtz Centre for Ocean Research, Kiel, Germany. [6] Institute of Space and Atmospheric Studies, University of Saskatchewan, Saskatoon, Saskatchewan, Canada. [7] Centrum für Erdsystemforschung und Nachhaltigkeit (CEN), Universität Hamburg, Hamburg, Germany. ✉email: juergen.bader@mpimet.mpg.de

There is general scientific agreement that global near-surface cooling prevailed during the common era until late in the nineteenth century[1,2]. However, controversy exists regarding whether the Holocene was characterized by long-term global cooling or warming. The first reconstruction of Holocene global annual temperatures[3] (hereafter referred to as Marcott reconstruction) showed long-term cooling during the late Holocene after the Holocene Thermal Maximum (HTM) (about 8–3 ka years before the common era (BCE)) (Fig. 1; black curve). This reconstructed global cooling trend resulted from strong trends in the marine temperature proxies selected from the North Atlantic sector[4]. In fact, the computed cooling would have not been apparent without inclusion of proxies from that region[4]. No physical explanation for the cooling trend has been given so far. In contrast to the Marcott reconstruction, a recent study[4] based on fossil pollen from North America and Europe (hereafter referred to as Marsicek reconstruction) suggests long-term warming during the Holocene until around 3000 BCE. The period afterwards shows no substantial millennia-long global mean temperature trend till the start of the CE after which global cooling occurs[4] (Fig. 1, red curve). The Marsicek reconstruction is in closer agreement with the long-term warming trend simulated by climate models[5], including our simulation shown in Fig. 1 (gray, green, blue curves). The warming in the simulations of Liu et al.[5] is caused by retreating ice sheets and increasing atmospheric greenhouse-gas concentrations.

The disagreement between the expected global warming from increasing greenhouse gases and retreating ice sheets and the cooling shown by the Marcott reconstruction is called the 'Holocene temperature conundrum'[5]. Hypotheses have been proposed to explain the conundrum. Marine proxies of sea-surface temperatures could be biased towards seasonal temperatures, and thus record orbitally driven changes in the seasonal cycle, while the annual mean temperature change might be small[6]. The computed mean out of a limited number of proxies with restricted spatial coverage might not be representative of the global mean.

Climate evolution during the last deglacation and the Holocene has been investigated in transient climate simulations. For example, Timm and Timmermann[7] use an intermediate complexity model to investigate the climate response to time-varying glacial-interglacial boundary conditions in a simulation of the last 210,00 years. Using a coupled ocean-atmosphere general circulation model a transient simulation of the climate evolution from the Last Glacial Maximum (about 19 BCE) to the Bølling-Allerød warming (about 12.5 BCE) was done by Liu et al.[8]. Smith and Gregory[9] performed transient climate runs simulating the last glacial cycle (the last 120 kyr) using a full atmosphere-ocean general circulation model, albeit at coarse resolution with the acceleration of climatic forcing by a factor of 10. Climate models exhibit substantial deficiencies, e. g. they might not correctly simulate the response to changes in radiative forcing (inaccurate climate sensitivity), processes are parameterized because of the relative coarse resolution, neglect some feedback mechanisms or might not incorporate these processes correctly—such as convection parameterisation, the cloud feedback, the migration of the Arctic treeline or changing atmospheric dust fluxes at low latitudes[6,10–12]. These biases influence not only regional climate parameters, but also the global mean temperature. A robust knowledge of our past temperature evolution requires a better understanding of the discrepancy of long-term temperature trends between individual proxy records and of model-data inconsistency.

Here, we contribute to such understanding by showing that a spatio-temporal analysis of annual temperature variability during the Holocene using data from a transient high-resolution (about

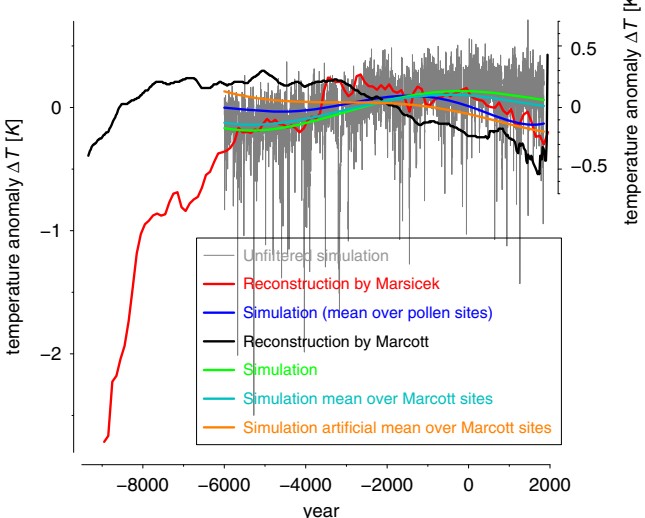

**Fig. 1 Annual global-mean temperature anomaly during the Holocene based on reconstructions and a transient simulation.** The annual global mean temperature anomaly based on the reconstructions by Marcott[3] (black curve), based on the reconstruction by Marsicek[4] (red curve), based on the transient unfiltered simulation (gray curve), based on the low-pass filtered simulation (green curve), based on the low-pass filtered simulation over the pollen sites used for the Marsicek reconstruction (blue curve), based on the low-pass filtered simulation over the Marcott sites (cyan), and based on an artificial simulation dataset over the Marcott sites (orange). All time series have been adjusted to the simulation time period (6000 BCE to 1850 CE). The units are Kelvin. The y-axis on the left side also shows the temperature anomaly but with a finer scale for easier comparison of the curves over the last millennia. For the computation for the artificial simulation dataset please see text for details.

200 km) climate model reveals that both a global warming mode and a cooling mode coexisted. We show that the sum of these two modes explains the long-term development in the global mean temperature and provide a physical mechanism for the cooling mode.

## Results

**The transient Holocene climate simulation**. We use the global spatio-temporal temperature variability in a transient Holocene simulation performed with the Max Planck Institute Earth System Model (MPI-ESM) spanning the period from 6000 BCE until 1850 CE. The model is forced by prescribed variations in the insolation (orbitally induced), greenhouse-gas concentrations, land-use changes, stratospheric volcanic aerosol distribution, spectral solar irradiance, and stratospheric ozone (Methods and Supplementary Figs. 1 and 2). The model accounts for dynamic vegetation changes in the land-surface model (see "Methods"). A new feature compared to previous transient Holocene simulations is that our transient Holocene simulation is additionally forced by solar variability and volcanic forcing. These additional forcing factors not only affect the interannual variability, but also have a substantial effect on the trend evolution of the global mean temperature by its cumulative/integrative effect.

**The warming and cooling mode**. The simulated centennial to millenial mean temperature variability during the Holocene is dominated by a warming and a cooling mode (Fig. 2) that together explain most of the long-term trends (explained variance about 99%; see Fig. 3b). We obtain these two modes by applying empirical orthogonal function (EOF) analysis to the modeled

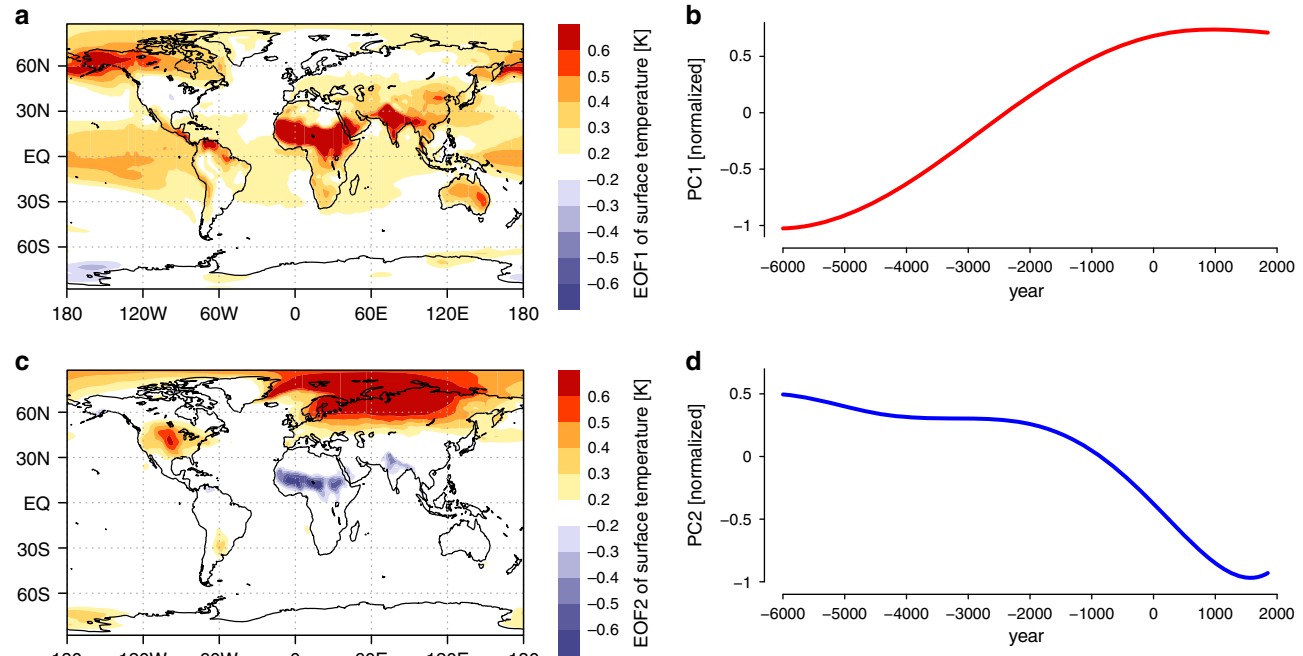

**Fig. 2 The simulated warming and cooling mode in the transient Holocene simulation.** First two spatial empirical orthogonal function (EOF) modes (a,c) and corresponding normalized smoothed principal components (PCs) (b,d) based on the simulated annual 2 m temperature using the MPI-ESM data. The red and blue curves in the right panels show the low-pass filtered PCs. The explained variances of the annual (not low-pass filtered) temperature modes are: 18% and 9%. Panels (a) and (b) show the spatial pattern and temporal evolution of the warming mode and panels (c) and (d) correspond to the cooling mode.

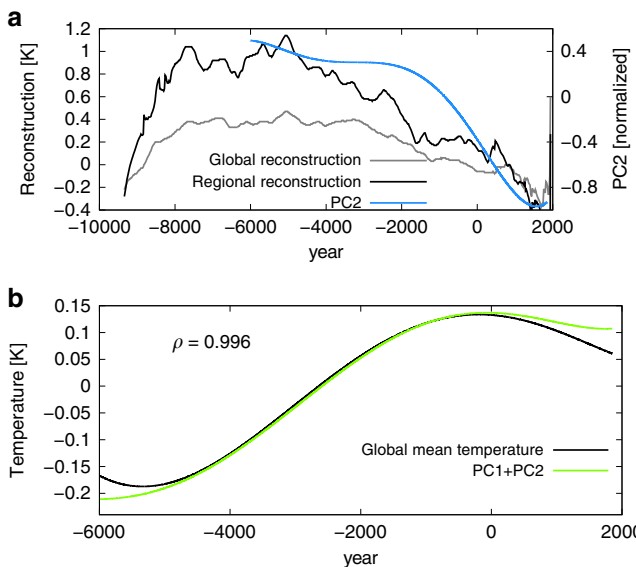

**Fig. 3 Net effect of the warming and cooling mode explains the long-term temperature trends during the Holocene. a** Temperature reconstructions based on Marcott et al. 2013 (gray and black curve) and the cooling mode (PC2) (blue curve). The gray curve shows the global mean temperature whereas the black curve shows a regional mean from 30°N poleward. **b** Low-pass filtered simulated annual global mean temperature anomaly (black curve) and sum of the weighted warming (PC1) and cooling (PC2) mode (green curve). The PCs have been weighted by the global means of the corresponding spatial patterns before summation. The global mean of EOF1 is about 0.238 K and of EOF2 it is about 0.066 K. $\rho$ is the correlation coefficient between the two curves.

annual temperature variability over the global domain. The global spatial patterns corresponding to the two modes have geographically well-separated centers of activity. The two spatial patterns show a clear separation between the tropics—in particular the monsoon regions—and the extratropical regions north of 60°N (the Arctic, Eurasia, and the North Atlantic). Furthermore, the temporal evolution of the two modes shows the strongest trends in different time periods. The long-term time evolution of the warming mode—defined as the smoothed first principal component (PC)—is dominated by a warming trend that is most pronounced over the first half (6000 BCE to 2000 BCE) of the simulation, whereas the smoothed second PC—trend evolution of the cooling mode—captures a substantial cooling tendency that is notably evident during the second half (see also Supplementary Fig. 3).

The EOF method decomposes the data into pairs of a spatial pattern and its associated time series. We call the spatial structure EOF and the corresponding time series PC. The greenhouse effect (Fig. 4a) is calculated as the difference between the upward surface thermal radiation and the outgoing longwave radiation at the top of the atmosphere. The increase in the greenhouse effect goes hand in hand with the warming shown by PC1. The CO2 increase happens mainly during the first half of the simulation and then flattens out (Supplementary Fig. 1b). The temporal PC1 weighted by the corresponding spatially averaged EOF represents the largest contribution to the simulated annual global mean temperature. The simulated global mean warming by the MPI-ESM is in agreement with previous studies. A robust global annual mean warming in the Holocene was simulated by climate models mainly in response to $CO_2$[5].

The spatial pattern (EOF1) of the simulated Holocene warming mode differs substantially from the warming pattern seen in future climate scenarios. For example, in contrast to the projected

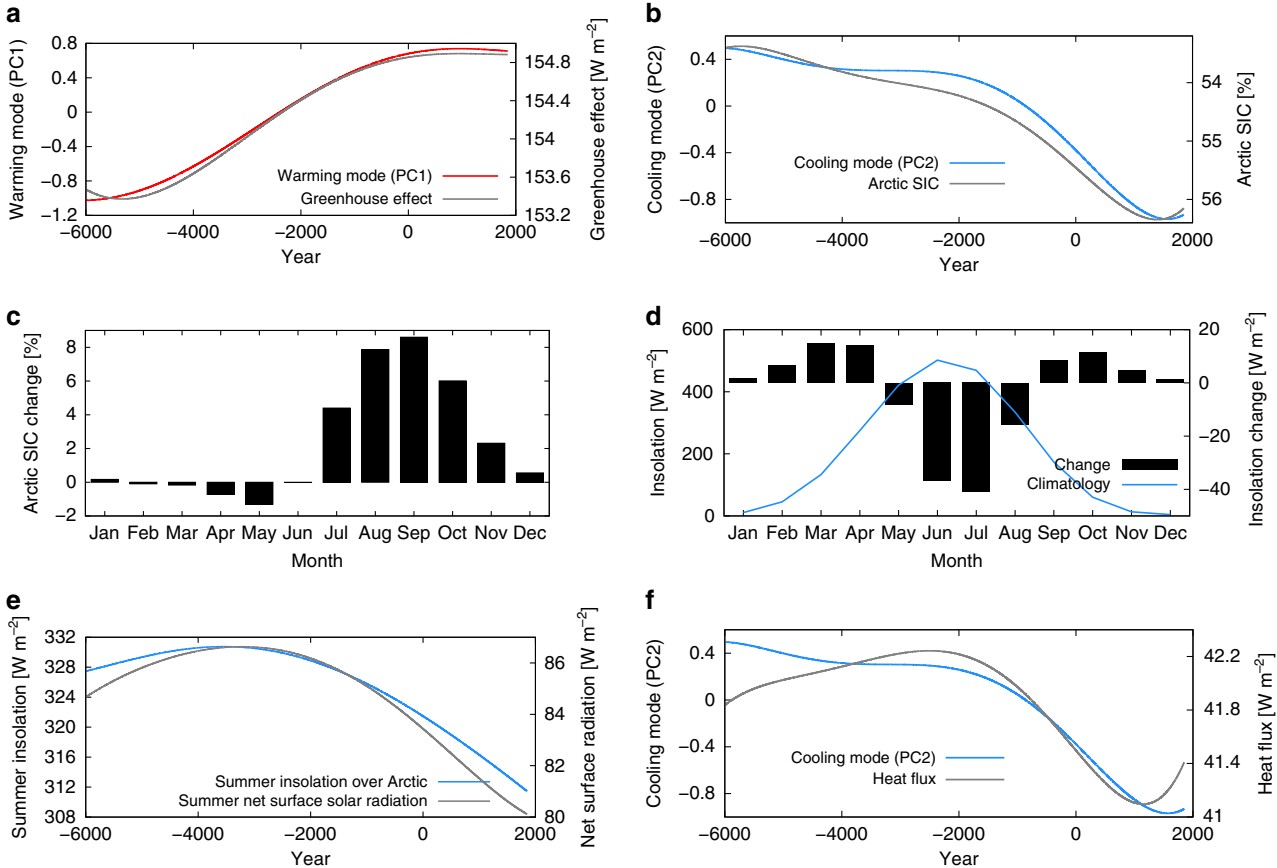

**Fig. 4 Time evolution of the warming mode is correlated with the greenhouse effect while the cooling mode is associated with the Arctic sea-ice increase. a** The gray curve shows the low-pass filtered greenhouse effect and the red curve the low-pass filtered warming mode (PC1). The greenhouse-gas effect is defined as the difference between the upward surface thermal radiation and the outgoing longwave radiation at the top of the atmosphere [W m⁻²]. **b** Low-pass filtered normalized cooling-mode (PC2) and low-pass filtered annual Arctic sea-ice concentration (SIC) in %. Please note that the right *y*-axis is flipped. **c** Change in the seasonal cycle of the Arctic SIC between the first and last one hundred years of the simulation. **d** Blue curve shows the simulated climatological insolation. Bars indicate the change in the seasonal cycle of the insolation between the first and last one hundred years of the simulation. **e** Low-pass filtered summer (JAS) insolation (blue curve) and summer (JAS) net surface solar radiation (black curve). **f** The gray curve shows the area-averaged heat flux from the ocean to the atmosphere in W m⁻² and the blue curve the cooling mode (PC2). Except for the warming mode (PC1), the cooling mode (PC2) and the greenhouse effect all data is area-averaged from 60° poleward.

future changes the Holocene warming mode is most pronounced in the tropical regions and hardly shows any Arctic amplification. This difference in the spatial warming pattern during the Holocene is related to the latitude-dependent trend in annual mean insolation forced by orbital changes. Although the global mean annual insolation does not change substantially, strong regional changes exist. For instance, the annual mean incoming solar radiation in the Arctic decreases, whereas the radiation in the tropics increases (Supplementary Fig. 1a). Moreover, globally, the strongest warming occurs in the West African and Indian monsoon regions, because of a positive feedback associated with the reduction in monsoonal rainfall and an associated decrease in evaporative cooling and total cloud cover.

The second mode (Fig. 2c, d) is a cooling mode. Its center of action is in the Northern Hemisphere extratropics approximately north of 60°N in the Barents Kara Sea, Eurasia north of 60°N, and the North Atlantic Ocean (Supplementary Notes 1 and Supplementary Figs. 4 and 5). Hence, the second mode appears most pronounced in the Arctic due to regional forcing variations and feedbacks. Increasing Arctic sea-ice cover leads to substantial cooling over the Arctic region and surrounding continents (Fig. 2c, Fig. 4b). Significant reduction in areal coverage and thickness of sea ice in the Arctic in response to increased summer and annual mean radiation at 7000 BCE compared to present was found in previous

modeling studies[13,14]. In addition, a significant mid-Holocene warming of the northern continents in summer is simulated in these experiments. Decreasing summer insolation from the mid-Holocene to the late Holocene causes an increase in Arctic sea-ice cover, in particular during late summer and early autumn, because of a reduction in summer Arctic sea-ice melt (Fig. 4c–e). The increasing Arctic sea-ice cover has two main effects that lead to near-surface cooling. First, it reduces the incoming solar radiation at the surface because of an enhanced surface albedo (Fig. 4e). The albedo cooling effect is strongest during late summer and early autumn because the large albedo enhancement goes along with relatively strong insolation during these months (Fig. 4d). Previous studies have demonstrated that changes in precession have an influence on the annual mean high latitude absorbed insolation and the sea-ice coverage, because of changes in the timing of seasonal insolation and its interactions/feedbacks with the sea-ice cover[15]. Second, the ocean in the high latitudes warms the atmosphere less because of the increased Arctic sea-ice-cover. The sea ice between the atmosphere and the ocean has an insulating effect by reducing the latent and sensible heat fluxes from the ocean to the atmosphere[16,17]. Therefore, the larger Arctic sea-ice cover in late summer and early autumn reduces the surface heat fluxes and thereby cools the atmosphere (Fig. 4f). The change in the seasonality from the mid-Holocene to the late Holocene not only

dominates the annual signal, but also prevails on the millennial time-scale in the Arctic region. The course of the summer Arctic insolation change from the mid-Holocene to the late Holocene imposes the time evolution of the cooling mode (see Fig. 4 b, e). The cooling mode is substantially amplified in our simulation that includes volcanic and solar irradiance forcing (Supplementary Fig. 6). The most prominent effect comes from volcanic eruptions and previous studies[18,19] have shown that clusters of eruptions can lead to long-lasting Northern Hemisphere cooling sustained by sea-ice/ocean feedbacks.

The time evolution of the Marcott reconstruction[3] shows more similarity to the cooling mode than to the warming mode. This raises the question whether the Marcott reconstruction is more a representative for the cooling mode rather than a proxy for the global mean temperature because of the proxy locations.

Therefore, we compute the simulated global mean temperature based only on the Marcott sites (Fig. 1, cyan curve). This model-based reconstruction shows a similar temperature evolution as the global mean simulated temperature. Therefore, we test the hypothesis that the marine proxies of sea-surface temperatures are biased towards the monthly maximum in a year. Following this idea, we construct an artificial temperature reconstruction based on the model data. For ocean points, we choose the monthly maximum of a year and for the land points we take the annual mean values. Using this artificial dataset the global mean over the Marcott sites shows a cooling similar to our cooling mode (Fig. 1, orange curve) and the cooling in the Marcott reconstruction. Therefore, our results not only confirm that the marine proxies of sea-surface temperatures used for the Marcott reconstruction might be biased towards the summer season, but also that at least during the last 3000 years a global cooling is in line with physical reasoning.

Under the assumption of a perfect model and correct forcing, the transient climate simulation suggests that computing the global mean temperature only over the pollen sites used by Marsicek[4] might overestimate the cooling tendency during the last two thousand years and underestimate the warming from 6000 BCE to 0 CE (see blue and green curves in Fig. 1).

## Discussion

Our study highlights that two modes associated with global warming and global cooling have presumably coexisted during the Holocene. Both modes are geographically well separated and have likely dominated the temperature evolution during different time periods in the Holocene. The multi-millenial temperature trends during the Holocene can be separated into these two modes. Moreover, the physical cause of the cooling mode is the time evolution of the change in the Arctic summer insolation via its influence on the Arctic sea-ice cover.

Interestingly, a substantial influence of higher frequency variability of the external forcing—volcanic activity and changes in the solar irradiance—in our model on the trend development of the global mean temperature exists. Supplementary Fig. 8 shows the global mean temperature for a similar transient simulation except that the volcanic forcing and changes in the solar irradiance are excluded. The simulation also has a warming and cooling mode spatial pattern similar to the patterns shown in Fig. 2 a and c. The striking difference in the global mean temperature is that the cooling during the last 3000 years is substantially reduced—compared to the simulation driven additionally with volcanoes and solar irradiance changes. This indicates that the cooling mode might integrate the higher variability (noise) of the external forcing in the transient climate simulation. The difference in the forcing therefore explains at least partly the important role of the cooling mode in our transient Holocene simulation.

Although we have demonstrated a substantial influence of the cooling mode on the global mean temperature, differences still exists between our simulated global mean temperature and the Marcott reconstruction (Fig. 1). In contrast to the Marcott reconstruction the warming dominates about the first half of the transient simulation. Further analyses support the fact that the marine proxies of sea-surface temperatures used for the Marcott reconstruction might be seasonally biased.

In addition to the MPI-ESM simulations, we have analyzed two more transient Holocene simulations using two different climate models (Supplemenatry Notes 2 and Supplementary Figs. 7 and 8). Although details of the warming pattern differ, all three models show a global warming mode indicating the robustness of the global warming signal in all three models. Only one (CCSM3) of the two additional models simulates the cooling mode. The inability of the FAMOUS model to simulate a cooling mode might be related to the high latitude cold bias[9] in the northern hemisphere. Nevertheless, this may also indicate some uncertainty in the cooling mode and in our conclusions.

## Methods

**Model and experimental design.** The Earth System Model of the Max Planck Institute for Meteorology (MPI-ESM1.2[20]) is used to perform the transient Holocene simulation spanning the time period from 6000 BCE until 1850 CE. Here, the atmospheric component ECHAM6[21] is run with a T63 horizontal resolution (1.875° × 1.875°) and 47 vertical levels. The ocean component MPI-OM[22] has a nominal horizontal resolution of 1.5° and applies 40 unevenly spaced vertical levels and includes the sea-ice model. The MPI-ESM also includes a dynamic vegetation model in the land-surface model JSBACH[23]. The transient simulation is forced by prescribed variations in the insolation (orbitally induced), in greenhouse-gas concentrations, land-use changes, stratospheric volcanic aerosol distribution, spectral solar irradiance, and stratospheric ozone. Changes of the ice sheets topography are not included in the transient simulation. The transient Holocene simulation is continued from a spinup experiment which has simulated over more than 3000 years. The spinup experiment was run with constant boundary conditions at 6k BCE. The climate model uses the Proleptic Gregorian calendar. In the following, a more detailed description of the boundary conditions and the forcing of the transient simulation are given.

The transient Holocene simulation is forced by prescribed variations in the orbital insolation, greenhouse-gas concentrations, land-use, stratospheric volcanic aerosol distribution, spectral solar irradiance and stratospheric ozone. The orbital forcing follows Berger[24]. Methane ($CH_4$), nitrous oxide ($N_2O$), and carbon dioxide ($CO_2$) concentrations are based on a new collection by Joos (personal communication). It is an updated version of the one used in Schmitt et al.[25]. The other forcing factors are described in more detail in individual paragraphs (see below). Supplementary Fig. 1 gives an overview of the individual boundary forcing factors.

**Solar forcing.** The time series of the solar total (TSI) and spectral (SSI) irradiance have been constructed using the SATIRE (Spectral And Total Irradiance Reconstructions) model[26]. SATIRE attributes irradiance changes to the continually evolving solar magnetism. The magnetic field emerges at the surface in form of dark (sunspots) and bright (facular and the network) features and modulates solar irradiance on time scales longer than roughly one day[27]. Using solar observations, SATIRE decomposes the visible solar surface into these various structures. Their brightness is considered to be time-independent and is pre-calculated from the appropriate solar model atmospheres[28]. Direct measurements of the solar surface magnetic field (solar full-disc magnetograms) are used since 1974[29]. Prior to that period, the reconstructions have to rely on proxies of solar magnetic activity.

The longest record of direct observations of solar activity is the sunspot number. It goes back to 1610 although the temporal coverage and the quality of the earlier data are relatively poor. Furthermore, the cross-calibration of the records by individual observers is an issue, and has been under intense debate recently[30–32]. Therefore, the reconstruction used here[33,34] employs the sunspot number[35] only in the period 1850–1974, where the differences between the various existing sunspot composites are negligible.

For the entire period prior to 1850 we have used radionuclide data. Since the production of the cosmogenic radionuclides in the terrestrial atmosphere is modulated by the solar magnetic activity, this allows a reconstruction of the past changes in the solar modulation potential, the solar open magnetic field, the sunspot number and finally the irradiance[34,36,37]. We used the $^{14}C$ data[38,39], since $^{14}C$ is mixed globally and is thus less affected by local conditions as compared to $^{10}Be$. Note that the long-term changes shown by the two isotopes are generally in good agreement with each other[40–45]. The cosmogenic isotope data have a decadal

temporal resolution. Therefore, the 11-year solar cycle has been simulated separately employing statistical relationships between various properties of the activity cycles and the decadally averaged sunspot numbers derived from direct sunspot observations[46].

The final reconstruction is normalized such that the absolute TSI level during the most recent activity minimum in 2008 matches SORCE/TIM measurements $(1360.8 \text{ W m}^{-2})$.

**Ozone forcing**. Stratospheric ozone varies with solar irradiance. The first-order effect is an increase of ozone in the upper stratosphere for higher UV irradiance due to the increased photodissociation of molecular oxygen ($O_3$). This increase is of the order of about 3% between typical maxima and minima of the 11-year solar cycle. This ozone effect enhances the stratospheric temperature response to solar irradiance variability[47] and may impact atmospheric dynamics down to the troposphere[48]. As the MPI-ESM does not treat stratospheric chemistry interactively, we prescribe monthly mean ozone ($O_3(t)$) in our simulations:

$$O_3(t) = O_{3,\text{CMIP5}} + \Delta I_{\text{UV}}(t) \times \frac{\partial O_{3,\text{CMIP6}}}{\partial I_{\text{UV}}}, \qquad (1)$$

where $O_{3,\text{CMIP5}}$ is the average monthly mean climatology used to force preindustrial CMIP5 simulations[49,50], $I_{\text{UV}}(t)$ is the monthly mean anomaly (calculated with respect to the average of years 1850–1860) of 200–320 nm solar irradiance taken from the dataset described above, and $\frac{\partial O_{3,\text{CMIP6}}}{\partial I_{\text{UV}}}$ are the monthly solar-ozone coefficients obtained by regression of an early version of the zonally averaged CMIP6 ozone dataset, which is based on simulations with two chemistry climate models for the period 1960–2011, to 200–320 nm solar irradiance[51]. $O_3(t)$ is nominally three dimensional, but like $O_{3,\text{CMIP5}}$ depends in the stratosphere only on latitude and altitude because only the tropospheric (up to $\approx 300$ hPa) part of $O_{3,\text{CMIP5}}$ depends on longitude while the stratospheric part and the solar-ozone coefficients are represented by zonal averages. All other potential sources of ozone variability, such as e.g. stratospheric volcanic aerosol or orbital forcing, are ignored in our simulations.

**Volcanic forcing**. Volcanic forcing is reconstructed from a 110 ka long record of volcanic sulfate from the Greenland ice core GISP2[52].

In contrast to reconstructions of volcanic forcing for the most recent millennia[53–56]—which use synchronized records from both Greenland and Antarctica—the use of a single core provides no information upon which to differentiate between eruptions of tropical and extratropical origin. We therefore made a rather strong simplifying assumption, taking each sulfur spike to represent a tropical eruption. Sulfate flux was scaled to volcanic stratospheric sulfur injection (VSSI) using transfer functions based on the 1991 Pinatubo eruption and analysis of radioactive debris from nuclear weapons testing[56]. Conversion of ice core-based eruptive histories into the aerosol properties needed by climate models is performed with the Easy Volcanic Aerosol module (EVA). EVA takes as input estimates of the VSSI (in Tg S), outputs wavelength-dependent aerosol extinction, single scattering albedo and scattering asymmetry factor values. Aerosol extinction is assumed to be linearly proportional to mass for eruptions smaller than the 1815 Tambora eruption, and follows a $\frac{2}{3}$ power-law scaling for larger eruptions[53]. An additional parametrization was implemented to accelerate the loss of sulfate aerosol mass for very large eruptions[57], with the parameterization based on model simulations of the Los Chochoyos eruption at approximately 84 ka BP[58].

The stratospheric aerosol optical depth (SAOD) reconstructed over the Holocene (Supplementary Figure 2), referred hereafter as the EVA(GISP2) reconstruction, shows reasonable agreement with reconstructions of the most recent centuries with a long-term annual mean global mean average of 0.017, 13% greater than that of the eVolv2k SAOD reconstruction for the past 2500 years[56]. This difference can be partially explained by the strong volcanic activity in the early Holocene recorded by the GISP2 sulfate record[52,59]. Due to non-linear parameterizations in the EVA forcing generator, the large sulfate events in the early Holocene produce SAOD peaks around 1.5 times that reconstructed for the 1257 Samalas eruption, but the magnitude of these forcings should be considered highly uncertain. Another very important uncertainty relates to periods of long duration elevated volcanic sulfate flux in the GISP2 sulfate record (e. g., around 5.2 ka BP). Zielinski et al.[60] proposed these to be artefacts due to elevated marine biogenic sulfur production, but they could also represent long lasting high latitude eruptions. Radiative forcing associated with these events in the EVA(GISP2) reconstruction should be considered an upper bound.

**Landuse forcing**. Landuse is based on the LUH2v1.0h product provided for the period after 850 AD for an earlier version of the dataset[61,62]. It includes wood harvest and land cover transitions that describe the expansion of cropland and pasture, but also their abandonment and shifting cultivation. To avoid an abrupt change in cropland and pasture area at 850 AD we introduce landuse starting at 150 BC by applying each year the same maps of relative transitions that lead after 1000 years to the cropland and pasture pattern of 850 AD. This results in an almost linear interpolation for most grid boxes as the share of agricultural land is small during this early period.

**Statistical methods and filtering**. Because we focus on the temperature trend during the Holocene, rather than at variations on shorter time-scales, all of the time series are smoothed by applying a polynomial fit of degree five so that it is a best fit—in a least-squares sense—for the data. The annual time series of the warming and cooling mode is also shown unfiltered and using a different smoothing (Supplementary Fig. 3). We applied an EOF analysis—also known as Principal Component Analysis (PCA)—to the global field of the annual mean temperature. The EOF analysis is a standard analysis technique in climate science to investigate spatial variability modes. The EOFs are obtained by computing the eigenvectors and eigenvalues of the spatially weighted covariance matrix of the temperature field. The first mode explains 18% and the second mode explains 9% of the interannual near-surface temperature variability. For the Marsicek and Marcott model temperature signal the nearest grid point to the proxy locations is picked. Afterwards an area-averaged global mean temperature over these selected grid points is computed using the simulated annual near-surface temperature data. We define an Early-Middle Holocene boundary at 8200 years BCE and a Middle-Late Holocene boundary at 4200 years BCE.

## Data availability
The data of the transient Holocene climate simulation performed with the MPI-ESM are available from the corresponding author upon request. All other data used in this study are available from public databases or literature, which can be found in the corresponding references. In particular, the Marcott data are available from the following link: "https://science.sciencemag.org/content/suppl/2013/03/07/339.6124.1198.DC1". The Marsicek data are available from the url: "https://www.nature.com/articles/nature25464#Sec21".

## Code availability
The code used in this study is available from the corresponding author upon request.

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

## Acknowledgements

The authors acknowledge the German Climate Computing Center (DKRZ) for providing the computer hardware for our model experiment. The research leading to these results has received funding from the German Federal Ministry of Education and Research (BMBF) through the JPI Climate CLIMPRE InterDec (FKZ: 01LP1609A) and PaCMEDy (FKZ:01LP1607B) projects and was done in the frame of the internal MPI-M project "Holocene". We also thank M. Kapsch and C. Reick for their helpful comments and feedback on the present manuscript.

## Author contributions

J.B. developed the essential research idea, carried out the analyses and wrote the first version of the paper. J.J. and M.C. contributed to developing the research. S.L. performed the climate simulation. N.K., A.M., T.R., H.S., M.T., and C.W. developed the forcing for the climate simulation. All contributed to the final version.

## Funding

## Competing Interests

The authors declare no competing interests.
