## [Peer Review File · Nature Communications]

Reviewer Comments, first round

Reviewer #1 (Remarks to the Author):

This paper has some interesting points. First, and most interestingly, it points out a cooling mode caused by the Arctic sea ice expansion in response to the decreased summer insolation in the Holocene. This cooling trend is represented in the EOF2 of Tair with the maximum loading in the high latitude region. Second, this paper relates this cooling mode to the Marcott reconstruction simply by exploring the similarity of the cooling trends between the Marcott global mean temperature and PC2. Finally, consistent with previous studies, they identified the EOF1 as the model forced by the increased CO2. Here, the first two points are novel. I would like to see a major revision of the paper.

Major comments:

1. The authors should also check if these two dominant modes are also present in the three simulations in ref. 5. If they do, which is most likely the case, they further show the robustness of their results and the significance of their modal decomposition idea will be enhanced significantly.
2. I don't quite follow the reasoning of point 2. PC2 is from mostly high latitude signal, while Marcott is global mean. A proper comparison should be made either in the model EOF2 space or in the proxy data space. In the model space, they should use a subset of Marcott proxy in the region consistent with EOF2 (high latitude...). Given, nevertheless, the cooling trend in the NH is even more clear than in global mean, as shown in Marcott, it is likely the cooling PC2 still resembles that in Marcott in the highlatitude region. So, their point 2 can still be valid, but with a more reasonable comparison. (Fig.4b should be replaced with regional mean of the proxy, in addition to the global mean). In the proxy data space, they could select the model temperature on the Marcott data sites, perhaps, also in the high latitude region, to see their similarity.

Minor:

1. What about their comparison with the regional temperature in Marsicek?
2. L93: what is the explained variance for PC1 and PC2 respectively? I can only find in the text their total is 99%. Also, what is the global mean value of each EOF? This will give us some sense of their potential contribution to the global mean..
3. L74: " ... dust fluxes at low latitudes". This may be relevant to a recent paper.

Liu et al. 2018: A possible role of dust in resolving the Holocene Temperature Conundrum. Scientific Report, 8, 4434, 10.1038/s41598-018-22841-5.

4. Figure. S2, S3: please explain in details how these effects are obtained from the set of sensitivity experiments. Write each effect as the difference between different pair of runs...As it stands now, I am somewhat confused, particularly on Fig.S3.
5. Fig.4d: the AOD effect is not discussed. A few sentences in the text, even speculative, on its role will be interesting. This is because this is a new effect that was not present in previous models, including those in ref. 5.

Reviewer #2 (Remarks to the Author):

The authors perform a transient simulation from 6,000 BCE using MPI-ESM. They then decompose the long-term 2-m temperature signals using EOF. From their analysis, Bader and others find two dominate modes of temperature variability: PC1 - a warming mode attributed to GHG increase, and PC2 - a cooling mode attributed to orbital change and sea ice feedback. The authors suggest

that these two modes can explain the divergent temperature trends found within the proxy reconstructions of Marcott et al. (2013) and Marsicek et al. (2018). Although the modeling component is strong, the limited analysis and lack of comparison with other works prevent this manuscript from fully answering the "Holocene Temperature Conundrum."

Main

Why the Marcott temperature dataset only records the secondary mode of variability is not well explained. Some additional analysis should be able to shed light on this outstanding ambiguity. Why not look at the seasonal signals further? Based on your argument, I think that you should be able to generally recreate both the Marcott and Marsicek temperature trend with some combination of seasonal data. If not, what does this suggest about the model results and/or the proxy data? The model data contain a lot more information than you currently utilize. Please take advantage of the model data. Also, a model-based temperature reconstruction using the Marcott sample sites should be added to Figure 1.

The manuscript is poorly referenced, especially given the amount of previous work on Holocene climate. Additional comparison with and acknowledgement of previous works is necessary. For example, how do your results compare with the other transient simulations of the Holocene (i.e. Liu et al., 2009; Smith et al., 2012; Timm et al., 2007)? Also, the proposed relationships between sea-ice and orbital change have been discussed in many previous works (e.g. Gallimore and Kutzbach, 1995; Tabor et al., 2014; Tuenter et al., 2005).

The lack of dynamic vegetation changes is a limitation that requires some discussion. For example, a Green Sahara, which present during much of the Holocene, has been shown to cause large-scale climate responses (e.g. Pausata et al., 2017; Swann et al., 2014). Further, a reduction in dust supply due to a Green Sahara was recently proposed to solve the Holocene Temperature Conundrum (Liu et al., 2018). Finally, vegetation changes resulting from orbital and CO₂ forcings can have significant impact on climate, especially in the high-latitudes (e.g. Claussen, 2009; Crucifix et al., 2002; Tabor et al., 2014; Tuenter et al., 2005). Although I do not think that you need to include a dynamic vegetation component in MPI-ESM, you should discuss the limitations that a lack of dynamic vegetation might introduce.

Other comments

43-44: "Marcott reconstruction" needs a citation.

70-71: When there are so many potential biases within Earth system models, it seems strange to highlight just climate sensitivity. Also, there are many papers on this topic, so please perform a more in-depth literature review.

74: Again, a lot of research beyond a basic nature summary (Shakun, 2018) has been done on this topic.

94-95: You use seasonal changes to explain your annual temperature signals yet you only perform EOF analysis on the annual signals...Worth exploring seasons?

114-115: This explanation is a bit simplistic.

116: Of course the change in annual incoming insolation is minimal.

125: "Amplification" should be lowercase.

135-136: Are you sure that it mainly a reduction in evaporative cooling? What about clouds?

144-146: Could some of this signal be an artificial calendar effect? I suspect that the contribution is minimal, but maybe not.

154-155: One of many studies to find this result. See above.

166: "...that, first, the..." strange wording

168-169: Does this need to be stated?

170-174: Or there are model biases or missing model components....

182-183: Please review previous literature more thoroughly.

243-244: A lot of the variance is unexplained by these modes. Is this a concern?

220: How did you initialize your simulation? Did you spin up the ocean and the land?

Figure 1: There seems to be extreme (annual?) variability in the raw data. Is this realistic? How did you create the Marsicek model temperature signal?

Figure 2: Display the amount of explained variance on the figure. Maybe reverse the color scale for panel (c)? Why are you using 2-m temperature? Is surface temperature more appropriate?

Supplemental

It is clear that the model setup received significantly more attention than the model analyses.

You discuss the O3 and solar irradiance forcings in depth, but these contribute minimally to your long-term 2-m temperature signal. Likewise, you discuss the land use forcing file in detail but is not very important for the results of your paper. I would much rather see a discussion about how the absence of dynamic vegetation might impact your results.

21: No reference for the CO2 forcing...?

Why didn't you consider both hemispheres for your volcanic forcing data?

Figure S1: Did you try your analysis with "a 101-year running mean"? Also, please differentiate the line colors.

Figure S2 and S3: How long were these experiments run for? What is the TOA imbalance?

References

Claussen, M., 2009. Late Quaternary vegetation-climate feedbacks. *Climate of the Past*, 5, pp.203-216.

Crucifix, M., Loutre, M.F., Tulkens, P., Fichefet, T. and Berger, A., 2002. Climate evolution during the Holocene: a study with an Earth system model of intermediate complexity. *Climate Dynamics*, 19(1), pp.43-60.

Gallimore, R.G. and Kutzbach, J.E., 1995. Snow cover and sea ice sensitivity to generic changes in Earth orbital parameters. *Journal of Geophysical Research: Atmospheres*, 100(D1), pp.1103-1120

Liu, Y., Zhang, M., Liu, Z., Xia, Y., Huang, Y., Peng, Y. and Zhu, J., 2018. A Possible Role of Dust in Resolving the Holocene Temperature Conundrum. *Scientific reports*, 8(1), p.4434.

Liu, Z., Otto-Bliesner, B.L., He, F., Brady, E.C., Tomas, R., Clark, P.U., Carlson, A.E., Lynch-Stieglitz, J., Curry, W., Brook, E. and Erickson, D., 2009. Transient simulation of last deglaciation with a new mechanism for Bølling-Allerød warming. *Science*, 325(5938), pp.310-314.

Liu, Z., Zhu, J., Rosenthal, Y., Zhang, X., Otto-Bliesner, B.L., Timmermann, A., Smith, R.S., Lohmann, G., Zheng, W. and Timm, O.E., 2014. The Holocene temperature conundrum. *Proceedings of the National Academy of Sciences*, 111(34), pp.E3501-E3505.

Pausata, F.S., Zhang, Q., Muschitiello, F., Lu, Z., Chafik, L., Niedermeyer, E.M., Stager, J.C., Cobb, K.M. and Liu, Z., 2017. Greening of the Sahara suppressed ENSO activity during the mid-Holocene. *Nature communications*, 8, p.16020.

Smith, R.S. and Gregory, J., 2012. The last glacial cycle: transient simulations with an AOGCM. *Climate dynamics*, 38(7-8), pp.1545-1559.

Swann, A.L., Fung, I.Y., Liu, Y. and Chiang, J.C., 2014. Remote vegetation feedbacks and the mid-Holocene green Sahara. *Journal of Climate*, 27(13), pp.4857-4870.

Tabor, C.R., Poulsen, C.J. and Pollard, D., 2014. Mending Milankovitch's theory: obliquity amplification by surface feedbacks. *Climate of the Past*, 10(1), pp.41-50.

Timm, O. and Timmermann, A., 2007. Simulation of the last 21 000 years using accelerated transient boundary conditions. *Journal of Climate*, 20(17), pp.4377-4401.

Tuenter, E., Weber, S.L., Hilgen, F.J. and Lourens, L.J., 2005. Sea-ice feedbacks on the climatic response to precession and obliquity forcing. *Geophysical research letters*, 32(24).

1 Response to the reviewers of the manuscript

2 “Global temperature modes shed light on the

3 Holocene temperature conundrum”

4 Jürgen Bader, Johann Jungclaus, Natalie Krivova,

Stephan Lorenz, Amanda Maycock, Thomas Raddatz,

Hauke Schmidt, Matthew Toohey, Chi-Ju Wu, Martin Claussen

May 4, 2019

Response to the reviewer #1:

*This paper has some interesting points. First, and most interestingly, it*
*points out a cooling mode caused by the Arctic sea ice expansion in response*
*to the decreased summer insolation in the Holocene. This cooling trend is rep-*
*resented in the EOF2 of Tair with the maximum loading in the high latitude*
*region. Second, this paper relates this cooling mode to the Marcott recon-*
*struction simply by exploring the similarity of the cooling trends between the*

*Marcott global mean temperature and PC2. Finally, consistent with previous*
*studies, they identified the EOF1 as the model forced by the increased CO2.*
*Here, the first two points are novel. I would like to see a major revision of*
*the paper.*

Thank you very much for your constructive comments.

Major comments:

1. *The authors should also check if these two dominant modes are also*
*present in the three simulations in ref. 5. If they do, which is most*
*likely the case, they further show the robustness of their results and the*
*significance of their modal decomposition idea will be enhanced signifi-*
*cantly.*

We were able to get the data of the CCSM3 (TraCe project) and FA-
MOUS (Quest project) model. Unfortunately, both models have a sub-
stantial Arctic sea ice concentration bias which might substantially in-
fluence the temperature modes. The Arctic sea ice concentration bias of
the FAMOUS model is e.g. documented in Smith and Gregory (2012):
“ ... FAMOUS has a high latitude cold bias in the northern hemisphere
during winter of about 5° C with respect to HadCM3 (averaged north
of 40°N), and a consequent overestimate of winter sea-ice extent in the

North Atlantic.” The overestimation of the sea-ice extent during the
period 1961 to 1990 in boreal winter (DJF) for the CCSM3 (TraCe)
simulation is shown in Figure 1 of this document. A substantial equa-
torward shift of the NH sea-ice extent is visible. The overestimation of
the sea-ice seems to be also reflected in the EOFs of the annual temper-
ature. The warming mode of the MPI-ESM simulation mainly shows
variability in the tropical regions in particular in the monsoon regions.
The strong variability in the monsoonal areas is also seen in the warm-
ing mode (EOF1) in the TraCe and Quest simulations (Figure 3 and
4 in this document). But in contrast to the MPI-ESM simulation the
warming mode in the TraCe and Quest simulations show substantial
temperature variability in the north Atlantic sector at the sea-ice edges.
This variability might therefore be an artefact of the too southward ex-
tend of the Arctic sea-ice edge in these two simulations. A similar
“cooling” mode as in the MPI-ESM simulation is visible in EOF3 of
the CCSM3 and FAMOUS simulations. In the TraCe simulation the
cooling mode shows a similar time behaviour as in the MPI-ESM sim-
ulation. In contrast, the Quest simulation shows not a cooling but
an associated warming of this pattern. We can only speculate about
these differences in the time behaviour. The FAMOUS model shows
substantial sea-ice variability (decreasing sea ice) at the west coast of

Figure 1: **Compared to observations the simulation shows in particular too high sea ice concentrations in the Arctic and an artificial equatorward extension of the sea ice edge in the North Atlantic and North Pacific.** Climatological winter (DJF) sea ice concentration in the CCSM3 simulation of the TraCe project. The climatological sea ice concentration is the annual mean over the period 1961 to 1990. [%]

Norway that might influence this pattern and only weak variability in
 the Barents-Kara Sea.

Figure 2: **Substantial sea ice variability in the mid-latitudes in the north Atlantic Ocean in the TraCe simulation.** Standard deviation of the winter (DJF) sea ice concentration in the CCSM3 simulation of the TraCe project. The standard deviation is computed over the period -5999 BCE to 1850 CE. [%]

Figure 3: First three global spatial EOF modes (a,c,e) and corresponding normalised smoothed PCs (b,d,f) based on the simulated annual near-surface temperature using the TraCe data (CCSM3 model). The explained variances of the annual temperature modes are: 17%, 12%, 8%.

Figure 4: First three global EOF modes (a,c,e) and corresponding normalised smoothed PCs (b,d,f) based on the simulated near-surface temperature using the Quest data (FAMOUS model). The explained variances of the annual temperature modes are: 6%, 6%, 4%.

2. I dont quite follow the reasoning of point 2. PC2 is from mostly high lat-
itude signal, while Marcott is global mean. A proper comparison should
be made either in the model EOF2 space or in the proxy data space. In
the model space, they should use a subset of Marcott proxy in the region
consistent with EOF2 (high latitude). Given, nevertheless, the cooling
trend in the NH is even more clear than in global mean, as shown in
Marcott, it is likely the cooling PC2 still resembles that in Marcott in
the highlatitude region. So, their point 2 can still be valid, but with a
more reasonable comparison. (Fig.4b should be replaced with regional
mean of the proxy, in addition to the global mean). In the proxy data
space, they could select the model temperature on the Marcott data sites,
perhaps, also in the high latitude region, to see their similarity.

First, we have redone the EOF analysis using only high latitude (pole-
ward of 60°N) data from the transient MPI-ESM simulation. The
warming and cooling mode patterns using global or high latitude data
look quite similar for the high latitude region (Figure 5 in this docu-
ment). One difference is that the cooling mode becomes the dominant
mode using only high latitude data. This shows that the cooling mode
pattern is not an artefact of the orthogonality constrain of the EOF
analysis using global data. The loadings of the spatial patterns using
regional data are a bit higher compared to the global patterns. The

corresponding PCs of the first two modes show a similar low-frequency
time evolution. The trend of the warming and cooling mode using only
high latitude data are weaker compared to the global PCs – but please
note that the spatial patterns have higher loadings.

In Figure 4a of the manuscript (please see also figure 7 of this docu-
ment) we have added a reconstruction based on proxy data poleward
of 30°N for a better comparison. The time evolution between the two
reconstructions shown in Figure 4a of the manuscript are quite similar.
But the reconstruction based on regional data shows a stronger cooling
trend compared to the reconstruction based on global data.

Figure 5: Comparison between the first two leading EOFs of the annual temperature in the transient MPI-ESM simulation when global data are used (a,c) and when only high latitude (poleward of 60°N) data are used (b,d). The explained variances of the first two regional modes are 30% and 15%.

Figure 6: Comparison between the first two leading PCs of the annual temperature in the transient MPI-ESM simulation when global data are used (a,c) and when only high latitude (poleward of 60°N) data are used (b,d).

Figure 7: **Net-effect of the warming and cooling mode explains the long-term temperature trends during the Holocene.** (a) temperature reconstructions based on Marcott et al. 2013 (grey and black curve) and the cooling mode (PC2) (blue curve). The grey curve shows the global mean temperature whereas the black curve shows a regional mean from 30°N poleward. (b) low-pass filtered simulated annual global mean temperature anomaly (black curve) and sum of the weighted warming (PC1) and cooling (PC2) mode (green curve). The PCs have been weighted by the global means of the corresponding spatial patterns before summation. ρ is the correlation coefficient between the two curves.

Minor comments:

1. *What about their comparison with the regional temperature in Mar-*
*sicek?*

The warming (PC1) and cooling mode (PC2) in our simulation individ-
ually show a relative weak relationship to the Marsicek reconstruction.
Please see Figure 8 in this document. This might be related to the fact
that the highest loadings of EOF1 and EOF2 are in the tropics and in
the Arctic respectively. The relationship to the global mean tempera-
ture or the superposition is better (please see Figure 1 and Figure 4 in
the manuscript).

Figure 8: Temperature reconstructions based on Marsicek et al. 2018 (grey) and the warming (PC1) (red curve) and cooling mode (PC2) (blue curve).

2. *L93: what is the explained variance for PC1 and PC2 respectively? I*

*can only find in the text their total is 99%. Also, what is the global*
*mean value of each EOF? This will give us some sense of their poten-*
*tial contribution to the global mean..*

We have computed the EOF analysis using annual mean values. The
explained variances for the interannual values was mentioned in the
method section. We have now added that information in the corre-
sponding figure. Please note that the explained variance for the inter-
annual values is much lower. For the low-pass filtered data the correla-
tion between PC1 (warming mode) and the global mean temperature
is about 0.98 (explained variance 96%). The correlation between the
low-pass filtered PC2 (cooling mode) and the low-pass filtered global
mean temperature is about 0.69 (explained variance 48%). Please note
that the modes have likely dominated the temperature evolution dur-
ing different time periods in the Holocene. The global mean value of
EOF1 is about 0.238 K and for EOF2 it is about 0.066 K. We have
added these scaling factors in Figure 4 of the manuscript.

3. *L74: dust fluxes at low latitudes. This may be relevant to a recent*
*paper. Liu et al. 2018: A possible role of dust in resolving the Holocene*
*Temperature Conundrum. Scientific Report, 8, 4434, 10.1038/s41598-*
*018-22841-5.*

We have now cited the paper by Liu.

4. *Figure. S2, S3: please explain in details how these effects are obtained*
*from the set of sensitivity experiments. Write each effect as the dif-*
*ference between different pair of runsAs it stands now, I am somewhat*
*confused, particularly on Fig.S3.*

In addition to the transient simulation, three other time-slice experi-
ments are performed with the MPI-ESM to better disentangle the effect
of the CO_2 increase and the influence of the change in the orbital pa-
rameters from the beginning (6000 BCE) to the end (1850 CE) of the
transient simulation. Each sensitivity experiment is run for 1000 years.
Therefore, one Holocene sensitivity experiment is performed with the
orbital forcing and the greenhouse gas concentration at 6000 BCE. The
simulations is called “6000BCE”. A second simulation is done with the
same orbital forcing, but the atmospheric CO_2 concentration is en-
hanced by 20 *ppm*. This enhancement corresponds approximately to
the difference between the CO_2 concentration at the beginning (6000
BCE) and the end (1850 CE) of the transient simulation. We will refer
to this simulation as “6000BCE+20ppm”. The simulations “6000BCE”
and “6000BCE+20ppm” differ only in the CO_2 concentrations. A
third Holocene simulation is performed with the orbital forcing at

1850 CE but with the greenhouse gas concentrations (CO_2, CH_4, N_2O)
at 6000 BCE. The simulation is called “1850CEghg6000BCE”. Suppl.
Figure 4a shows the near-surface annual mean temperature difference
between the simulations “6000BCE+20ppm” and “6000BCE”. There-
fore it shows the temperature change caused by a 20 *ppm* CO_2 in-
crease. Suppl. Figure 4b shows the near-surface temperature differ-
ence between the simulations “1850CEghg6000BCE” and “6000BCE”.
The temperature change in the annual mean temperature is caused
by changing the orbital parameters at 6000 BCE to 1850 CE values.
Suppl. Figure 5 shows the difference in the greenhouse effect between
the simulations “1850CEghg6000BCE” and “6000BCE”.

5. *Fig.4d: the AOD effect is not discussed. A few sentences in the text,*
*even speculative, on its role will be interesting. This is because this is a*
*new effect that was not present in previous models, including those in*
*ref. 5.*

We have added the following text:”In contrast to previous transient
Holocene simulations our transient Holocene simulation is addition-
ally forced by solar variability and volcanic forcing. A first analysis
indicates that these additional forcing factors not only effect the in-
terannual variability, but also increase the multi-decadal to centennial

variability.”

Response to the reviewer #2:

*The authors perform a transient simulation from 6,000 BCE using MPI-*
*ESM. They then decompose the long-term 2-m temperature signals using*
*EOF. From their analysis, Bader and others find two dominate modes of*
*temperature variability: PC1 - a warming mode attributed to GHG increase,*
*and PC2 a cooling mode attributed to orbital change and sea ice feedback.*
*The authors suggest that these two modes can explain the divergent temper-*
*ature trends found within the proxy reconstructions of Marcott et al. (2013)*
*and Marsicek et al. (2018). Although the modeling component is strong,*
*the limited analysis and lack of comparison with other works prevent this*
*manuscript from fully answering the Holocene Temperature Conundrum.*

Thank you very much for your suggestions. We have done additional analysis
and have compared our findings with previous works.

Main

1. *Why the Marcott temperature dataset only records the secondary mode*
*of variability is not well explained. Some additional analysis should be*
*able to shed light on this outstanding ambiguity. Why not look at the*
*seasonal signals further? Based on your argument, I think that you*

*should be able to generally recreate both the Marcott and Marsicek tem-*
*perature trend with some combination of seasonal data. If not, what*
*does this suggest about the model results and/or the proxy data? The*
*model data contain a lot more information than you currently utilize.*
*Please take advantage of the model data. Also, a model-based temper-*
*ature reconstruction using the Marcott sample sites should be added to*
*Figure 1.*

We have done additional analysis. The model-based temperature re-
constructions using the Marcott sample sites were computed and added
to Figure 1 in the manuscript. This model-based reconstruction fails
to show a cooling - although the warming is weakened compared to
the global mean from all grid points. Therefore, we test the hypothesis
that the marine proxies of sea-surface temperatures are biased towards
the monthly maximum in a year. Following this idea, we constructed
an artificial temperature reconstruction based on the model data. For
ocean points we choose the monthly maximum of a year and for the land
points we take the annual mean values. Using this artificial dataset the
global mean over the Marcott sites shows a cooling similar to our cool-
ing mode (Figure 1 a,b in the manuscript, orange curves). Therefore,
our results indicate that the Marcott reconstruction is biased towards
the cooling mode and the marine proxies of sea surface temperatures

might be biased towards the monthly maximum in a year.

2. *The manuscript is poorly referenced, especially given the amount of*
*previous work on Holocene climate. Additional comparison with and*
*acknowledgement of previous works is necessary. For example, how*
*do your results compare with the other transient simulations of the*
*Holocene (i.e. Liu et al., 2009; Smith et al., 2012; Timm et al., 2007)?*
*Also, the proposed relationships between sea-ice and orbital change have*
*been discussed in many previous works (e.g. Gallimore and Kutzbach,*
*1995; Tabor et al., 2014; Tuenter et al., 2005).*

We have added the mentioned references and have put our findings into
the context of previous work.

3. *The lack of dynamic vegetation changes is a limitation that requires*
*some discussion. For example, a Green Sahara, which present during*
*much of the Holocene, has been shown to cause large-scale climate re-*
*sponses (e.g. Pausata et al., 2017; Swann et al., 2014). Further, a*
*reduction in dust supply due to a Green Sahara was recently proposed*
*to solve the Holocene Temperature Conundrum (Liu et al., 2018). Fi-*
*nally, vegetation changes resulting from orbital and CO₂ forcings can*
*have significant impact on climate, especially in the high-latitudes (e.g.*
*Claussen, 2009; Crucifix et al., 2002; Tabor et al., 2014; Tuenter et*

*al., 2005). Although I do not think that you need to include a dynamic*
*vegetation component in MPI-ESM, you should discuss the limitations*
*that a lack of dynamic vegetation might introduce.*

This is a misunderstanding. Our Earth system model includes dynamic
vegetation changes. We have made that more clear in the revised ver-
sion. The dynamic vegetation is included in the land surface model
JSBACH. Therefore, our results include the effect of dynamic vegeta-
tion changes.

Other comments

1. 43-44: *Marcott reconstruction needs a citation.*

done

2. 70-71: *When there are so many potential biases within Earth system*
*models, it seems strange to highlight just climate sensitivity. Also, there*
*are many papers on this topic, so please perform a more in-depth liter-*
*ature review.*

We have rewritten that part and added more citations.

3. 74: *Again, a lot of research beyond a basic nature summary (Shakun,*
*2018) has been done on this topic.*

We have rewritten that part and added more citations.

4. *94-95: You use seasonal changes to explain your annual temperature*
*signals yet you only perform EOF analysis on the annual signals. Worth*
*exploring seasons?*

It is definitely interesting to look into seasons. Partly we have done
this to recreate the Marcott temperature record. Please see your first
main point. Probably we will have a closer look at seasons in our future
work.

5. *114-115: This explanation is a bit simplistic.*

We wonder if we understand the reviewer correctly. We want to say that
in the transient MPI-ESM simulation a warming during the Holocene
is simulated mainly in response to increasing CO₂. We do not want to
exclude other factors like e.g. non-linear feedbacks, or other greenhouse
gases. Please note that our simulation does not account for the retreat
of ice sheets.

6. *116: Of course the change in annual incoming insolation is minimal.*

Please see previous point.

7. *125: Amplification should be lowercase.*

changed

8. *135-136: Are you sure that it mainly a reduction in evaporative cooling?*

*What about clouds?*

We agree that the statement is too strong. We checked the total cloud
cover. There is a substantial downward trend in the annual mean total
cloud cover. We have modified the sentence accordingly.

9. 144-146: *Could some of this signal be an artificial calendar effect? I*
*suspect that the contribution is minimal, but maybe not.*

We cannot rule out that some of this signal could be an artificial cal-
endar effect. Because we have only monthly data, we have redone the
calculation for JJAS to account at least partly for the calendar effect.

This computation indicates that the conclusions are robust.

10. 154-155: *One of many studies to find this result. See above.*

We have added another citation.

11. 166: *that, first, the strange wording*

*modified*

12. 168-169: *Does this need to be stated?*

We have removed this part of the the sentence.

13. 170-174: *Or there are model biases or missing model components.*

We have modified the text.

14. 182-183: *Please review previous literature more thoroughly.*

We have changed that sentence: “Moreover, the physical cause of the
cooling mode is the time evolution of the change in the Arctic summer
insolation via its influence on the Arctic sea-ice cover.”

15. *243-244: A lot of the variance is unexplained by these modes. Is this a*
*concern?*

We have used annual mean values for the EOF analysis. Therefore, this
is the explained variance for the interannual variability. The explained
variance for the low-pass filtered data is much higher. Therefore, we
think it is not a major concern.

16. *220: How did you initialize your simulation? Did you spin up the ocean*
*and the land?*

The transient Holocene simulation is continued from a spin-up exper-
iment which has simulated over more than 3000 years. The spin-up
experiment was run with constant boundary conditions at 6k BCE. We
have added this information in the Methods section.

17. *Figure 1: There seems to be extreme (annual?) variability in the raw*
*data. Is this realistic? How did you create the Marsicek model temper-*
*ature signal?*

It is likely that we overestimate the extreme variability because of too
strong volcanic forcing. This too strong volcanic forcing seems to be an

artefact of just using an ice-core from one hemisphere. We have con-
structed a new volcanic forcing that considers both hemispheres. The
experiments are running. For the Marsicek and Marcott model tem-
perature signal the nearest grid point to the proxy locations is picked.
Afterwards an area-averaged global temperature mean over these se-
lected grid points using the simulated annual near-surface temperature
data. This is repeated for every model year. We have added this infor-
mation to the Methods section.

18. *Figure 2: Display the amount of explained variance on the figure.*
*Maybe reverse the color scale for panel (c)? Why are you using 2-*
*m temperature? Is surface temperature more appropriate?*

We have added the amount of explained variance in the caption. We
would like to keep the color scale. This makes it possible to directly
compare e.g. the PC2 with the Marcott reconstruction without flip-
ping the sign. We have briefly compared the 2-m temperature and the
surface temperature. The differences are not large. We also recom-
puted the EOFs using the surface temperature. It does not change our
conclusions.

19. *It is clear that the model setup received significantly more attention than*

*the model analyses. You discuss the O3 and solar irradiance forcings*
*in depth, but these contribute minimally to your long-term 2-m tem-*
*perature signal. Likewise, you discuss the land use forcing file in detail*
*but is not very important for the results of your paper. I would much*
*rather see a discussion about how the absence of dynamic vegetation*
*might impact your results.*

Please see our comment to your major comment.

20. *21: No reference for the CO2 forcing?*

We have added a reference. It is an updated version of the one used in:
“Carbon Isotope Constraints on the Deglacial CO2 Rise from Ice Cores”
by Jochen Schmitt et al. in Science 2012

21. *Why didnt you consider both hemispheres for your volcanic forcing*
*data?*

Please see our former comment: “It is likely that we overestimate the
extreme variability because of too strong volcanic forcing. This too
strong volcanic forcing seems to be an artefact of just using an ice-core
from one hemisphere. We have constructed a new volcanic forcing that
considers both hemispheres. The experiments are running.”

22. *Figure S1: Did you try your analysis with a 101-year running mean?*

*Also, please differentiate the line colors.*

We have not performed an analysis with the 101-year running mean.

We have differentiated the line colours.

23. *Figure S2 and S3: How long were these experiments run for? What is*
*the TOA imbalance?*

Each experiment is run for 1000 years. We have added this information

in the supplement. The mean TOA imbalance is between 0.38 and 0.49

344 Wm^{-2} in the different experiments.

Reviewers' comments, second round:

Reviewer #1 (Remarks to the Author):

The authors have responded to my questions to some extent. Nevertheless, I don't think the authors have addressed the questions satisfactorily. Therefore, I recommend the authors to have another revision of the paper, especially to address my two major concerns.

1. The neglect of the discussion of other models.

For the EOFs in FAMOUS and TRACE, the authors did do additional analyses, but didn't discuss them in the text because, it seems to me, these models don't fit their model story. This assumes their model MPI-ESM is the only correct one. I don't feel this a useful way forward. There are certainly many things wrong in every model, including their model! Here, even with excessive sea ice, why the cooling mode is not reproduced well in FAMOUS? I would have thought the sea ice would amplify the high latitude feedback and therefore enhance the arctic cooling mode, if it is generated by this mechanism.

A more useful way, and more constructive way, I think, is to modify their conclusion to reconcile the analyses in other models. For example, all three models show the global warming mode as EOF1, although the detailed region in high latitudes may differ. Indeed, the warming loading is not limited in the tropics in MPI-ESM either: there is a significant warming over Alaska! In the other two models, the high latitude warming center is located over the North Atlantic, perhaps related to the excessive sea ice. All the three models therefore show that the dominant signal is global warming, as in their global mean annual mean! This is therefore a robust result in all models.

By the way, usually, in response to CO₂, there is a polar amplification. This warming mode, however, is not. The authors should explain why.

The 2nd mode, cooling mode in high latitude, seems to be also consistent with CCSM3, but not with FAMOUS. This may indicate some uncertainty in this mode and in turn their conclusion. In short, some discussions on other analyses of other models are necessary here.

2: What is the implication of this paper, or its contribution to the understanding of the conundrum problem?

For discussion purpose, let's assume MPI model is indeed correct and there are two modes, one warming and one cooling. But then, how does this help resolve the conundrum? Is it a data problem (e.g. seasonal bias) or model problem (e.g. failure to produce the cooling mode because of the lack of some feedback?)

The conclusion in L217 suggests that: "The agreement in the temporal progression between the simulated cooling mode and the Marcott reconstruction suggests that this reconstruction is more representative for the global cooling mode. Moreover, the physical cause of the cooling mode is the time evolution of the change in the Arctic summer insolation via its influence on the Arctic sea-ice cover." This seems to suggest that Marcott captures the cooling mode more than the warming mode. But, in their model MPI-ESM, there is no evidence supporting this. Indeed, the annual mean temperature on the Marcott site show a warming, not cooling, as shown now in Fig.1. This is the same as in other models.

On the other hand, in L203-, the paper addressed this problem, more specifically, "Therefore, our results indicate that the Marcott reconstruction is biased towards the cooling mode and the marine proxies of sea surface temperatures might be biased towards the monthly maximum in a year.". This seems to mix two things: the cooling mode is for annual mean, as shown in their model, while the seasonal bias is a data bias that has been discussed in ref. 5. My reading of the paper seems to suggest that the key problem of Marcott is still the seasonal bias. This paper therefore supports which the basic hypothesis of ref. 5, but it is not specified so it left it unclear what does it mean by "Marcott reconstruction is biased towards the cooling mode", is it in the annual mean or it is the seasonal bias? It seems to me, in the MPI-ESM model, the annual mean cooling mode is produced (assume the model is correct), but the global annual mean is still warming. So even if Marcott observation correctly produce the cooling mode, the global mean annual mean is still warming.

Therefore, the seasonal bias is critical here, as suggested in previous works. This needs to be clarified, because this is the key point of the paper.

Reviewer #2 (Remarks to the Author):

Overall, the authors have done a good job responding to my initial comments. Some of my concerns came from a misunderstanding of the model configuration. The additional method details included in this revised version should prevent similar confusion by future readers. I still think that the authors could provide additional citations of previous works on the topic.

Comments:

The TOA imbalance is high in some of these supplemental simulations. You should mention the TOA imbalances in the text.

You should mention the calendar you used and why the associated biases are likely of secondary importance. It still could be a significant issue for figure 3 c) and d).

So, crops are prescribed but everything else is dynamic? Out of curiosity, do you get a "green" Sahara? Does this influence dust? Generally, I think that the contribution from aerosols might be worth expanding.

I cannot read the yellow text in the figures.

I am not sure that you need figure 1 b).

The comparisons with FAMOUS and TraCE simulations is interesting. I think this comparison is worth mentioning in the main text and including in the supplemental material.

Old text for figure 3 is still in revised manuscript.

How will people be able to access the model data?

Reviewer #3 (Remarks to the Author):

This is the review of the revision of "Global temperature modes shed light on the Holocene temperature conundrum" by Bader et al.

The authors used MPI-ESM for both transient simulation of past 8000 years and three sensitivity experiments to investigate the Holocene temperature conundrum. The major finding is that a global warming mode and a cooling mode are the first two EOFs of annual temperature variability from the transient simulation, and the authors attribute the warming mode to the 20ppm CO₂ increase (and latitudinal trend of annual mean insolation) while the cooling trend to orbitally forced changes in the Arctic sea-ice.

Both the transient and sensitivity simulations were very well designed and I commend the authors for completing these long simulations.

But I don't think the authors can support their conclusion based on the evidence they provided.

My overall criticism for this paper is that there is too much inconsistency between the transient simulation and the two reconstructions that I can't see how the authors can relate the findings from the transient simulation to the two reconstructions.

In Figure 1, the authors show that the modeled global mean temperature failed to produce the global cooling from the Marcott reconstruction. In order to produce the cooling trend, the authors

had to follow what's been done in ref.5 and use summer SST instead of annual SST in the global temperature stack. But as pointed in ref.5, the seasonally biased stack "does not improve model-data inconsistency significantly across individual sites. The spatial correlation of the temperature trends across the 73 sites of Marcott stack between the data, and the ensemble model temperature for the seasonally biased stack (0.01) is not significantly different from that for the annual temperatures (0.16)". So I assume that the seasonal biased "simulation artificial mean" stack from MPI-ESM won't correct the model-data inconsistency significantly across individual sites.

In reply to reviewer 1, the authors acknowledge that "The warming (PC1) and cooling mode (PC2) in our simulation individually show a relative weak relationship to the Marsicek reconstruction."

Therefore, the transient simulation fails to catch the major characteristics of either reconstruction, which makes it not credible to relate the warming mode and cooling mode in the transient simulation to the mechanisms behind the trend in the proxy reconstructions.

Another criticism is related to the authors' attribution of warming mode to the 20ppm CO2 increase (and latitudinal trend of annual mean insolation). This is not correct and not consistent with the results of sensitivity simulations shown in supplemental Figure 4.

In fact, the warming mode of PC1 with maximum warming in Tropical Africa and India in Figure 2a is mostly due to the changing orbital parameters and the associated monsoon changes, which the authors have already demonstrated clearly in their sensitivity experiments in Figure S4. Figure S4a shows the pattern of the warming from greenhouse gases with the classic pattern of polar amplification, but doesn't show hardly any warming in the aforementioned monsoon region in Tropical Africa and India in the warming mode in Figure 2a. On the other hand, the warming due to changing the orbital parameters in Figure S4b clearly matches the maximum warming in Tropical Africa and India found in the warming mode of PC1 in Figure 2a, which is in the monsoon regions that is consistent with the monsoon changes from changing orbital parameters. Therefore, the sensitivity experiments clearly demonstrate that the warming mode in PC1 is due to changing the orbital parameters, not due to 20ppm CO2 increase (and latitudinal trend of annual mean insolation).

In summary, both the warming mode and the cooling mode are associated with the changes in orbital parameters, and EOF analysis of transient simulation shows that the first two leading mode of variability in the transient simulation are both due to the changes of orbital parameters.

Minor comments:

1. change "effect" to "affect" in L104
2. change "D. Shakun, J." to "Shakun, J. D."
3. Add a note on whether the change of the ice sheets topography is included in the transient simulation between 8000 years ago and 6000 years ago.

1 Response to the reviewers of the manuscript

2 “Global temperature modes shed light on the

3 Holocene temperature conundrum”

4 Jürgen Bader, Johann Jungclaus, Natalie Krivova,

Stephan Lorenz, Amanda Maycock, Thomas Raddatz,

Hauke Schmidt, Matthew Toohey, Chi-Ju Wu, Martin Claussen

October 16, 2019

Response to the reviewer #1:

*The authors have responded to my questions to some extent. Nevertheless, I*

*dont think the authors have addressed the questions satisfactorily. Therefore,*

*I recommend the authors to have another revision of the paper, especially to*

*address my two major concerns.*

Major comments:

1. *The neglect of the discussion of other models. For the EOFs in FA-*
*MOUS and TRACE, the authors did do additional analyses, but didnt*
*discuss them in the text because, it seems to me, these models dont*
*fit their model story. This assumes their model MPI-ESM is the only*
*correct one. I dont feel this a useful way forward. There are certainly*
*many things wrong in every model, including their model! Here, even*
*with excessive sea ice, why the cooling mode is not reproduced well in*
*FAMOUS? I would have thought the sea ice would amplify the high lat-*
*itude feedback and therefore enhance the arctic cooling mode, if it is*
*generated by this mechanism.*

*A more useful way, and more constructive way, I think, is to modify*
*their conclusion to reconcile the analyses in other models. For exam-*
*ple, all three models show the global warming mode as EOF1, although*
*the detailed region in high latitudes may differ. Indeed, the warming*
*loading is not limited in the tropics in MPI-ESM either: there is a sig-*
*nificant warming over Alaska! In the other two models, the high latitude*
*warming center is located over the North Atlantic, perhaps related to*
*the excessive sea ice. All the three models therefore show that the dom-*
*inant signal is global warming, as in their global mean annual mean!*

*This is therefore a robust result in all models.*

*By the way, usually, in response to CO₂, there is a polar amplification.*

*This warming mode, however, is not. The authors should explain why.*

*The 2nd mode, cooling mode in high latitude, seems to be also con-*
*sistent with CCSM3, but not with FAMOUS. This may indicate some*
*uncertainty in this mode and in turn their conclusion. In short, some*
*discussions on other analyses of other models are necessary here.*

Thank you very much for your suggestion. We have added the ad-
ditional analyses for the FAMOUS and TRACE simulations in the
supplement of the manuscript and the corresponding results are now
discussed in the main manuscript. We have added the following text
in the manuscript: “In addition to the MPI-ESM simulation we have
analysed two additional transient Holocene simulations using two dif-
ferent climate models (suppl. Figures 9 and 10). Although details
of the warming pattern differ all three models show a global warming
mode indicating the robustness of the global warming signal in all three
models. Only one (CCSM3) of the two additional models simulates the
cooling mode. This may indicate some uncertainty in the cooling mode
and in our conclusions.

Concerning the “missing/reduced” polar amplification in response to
CO₂ we have commented in the text: “Interestingly, the spatial pattern
of the simulated Holocene warming mode differs substantially from the
warming pattern seen in future climate scenarios. For example, in
contrast to the projected future changes the Holocene warming mode
is most pronounced in the tropical regions and hardly shows any Arctic
amplification. This difference in the spatial warming pattern during the
Holocene is related to the latitude-dependent trend in annual mean
insolation forced by orbital changes. Although the global mean annual
insolation does not change substantially, strong regional changes exist.
For instance, the annual mean incoming solar radiation in the Arctic
decreases, whereas the radiation in the tropics increases (suppl. Figure
1a). As a consequence the greenhouse effect – because it is temperature
dependent – is reinforced in the tropics and almost offset in the Arctic
(see also suppl. Figures 4,5).”

Please also see our comment to reviewer 3.

2. *What is the implication of this paper, or its contribution to the under-*
*standing of the conundrum problem?*

*For discussion purpose, lets assume MPI model is indeed correct and*
*there are two modes, one warming and one cooling. But then, how does*

*this help resolve the conundrum? Is it a data problem (e.g. seasonal*
*bias) or model problem (e.g. failure to produce the cooling mode because*
*of the lack of some feedback?)*

*The conclusion in L217 suggests that: The agreement in the temporal*
*progression between the simulated cooling mode and the Marcott recon-*
*struction suggests that this reconstruction is more representative for the*
*global cooling mode. Moreover, the physical cause of the cooling mode*
*is the time evolution of the change in the Arctic summer insolation*
*via its influence on the Arctic sea-ice cover. This seems to suggest that*
*Marcott captures the cooling mode more than the warming mode. But,*
*in their model MPI-ESM, there is no evidence supporting this. Indeed,*
*the annual mean temperature on the Marcott site show a warming, not*
*cooling, as shown now in Fig.1. This is the same as in other models.*
*On the other hand, in L203-, the paper addressed this problem, more*
*specifically, Therefore, our results indicate that the Marcott reconstruc-*
*tion is biased towards the cooling mode and the marine proxies of sea*
*surface temperatures might be biased towards the monthly maximum in*
*a year.. This seems to mix two things: the cooling mode is for annual*
*mean, as shown in their model, while the seasonal bias is a data bias*
*that has been discussed in ref. 5. My reading of the paper seems to*

*suggest that the key problem of Marcott is still the seasonal bias. This*
*paper therefore supports which the basic hypothesis of ref. 5, but it is*
*not specified so it left it unclear what does it mean by Marcott recon-*
*struction is biased towards the cooling mode, is it in the annual mean*
*or it is the seasonal bias? It seems to me, in the MPI-ESM model,*
*the annual mean cooling mode is produced (assume the model is cor-*
*rect), but the global annual mean is still warming. So even if Marcott*
*observation correctly produce the cooling mode, the global mean annual*
*mean is still warming. Therefore, the seasonal bias is critical here, as*
*suggested in previous works. This needs to be clarified, because this is*
*the key point of the paper.*

We have modified the text to make clearer what the implications of
this paper are and what its contribution to the conundrum problem is.

In particular we modified the conclusion.

We think our manuscripts contains several new aspects:

- • we show for the first time that a climate model is able to simulate
a global cooling mode during the holocene
- • we highlight the processes involved in producing the cooling mode
- • in the revised version we make clear that the volcanic forcing am-
plifies the cooling mode

- • we show that a substantial influence of higher frequency variability
of the external forcing on the trend development of the global
mean temperature exists by integration of these effects
- • the warming and cooling mode computed from interannual values
define more or less the global mean temperature trend evolution

Although we have a substantial influence of the cooling mode on the
global mean temperature, differences still exists between our simulated
global mean temperature and the Marcott reconstruction (Figure 1). In
contrast to the Marcott reconstruction the warming dominates about
the first half of the transient simulation. Further analysis – using an
artificial temperature reconstruction – confirms that the Marcott re-
construction might have been biased towards the summer (Figure 1;
orange curve). Nevertheless, we show that a global cooling mode exists
in our model and we give a physical explanation for it. Please also see
the comment to reviewer3.

Response to the reviewer #2:

*Overall, the authors have done a good job responding to my initial com-*
*ments. Some of my concerns came from a misunderstanding of the model*
*configuration. The additional method details included in this revised version*
*should prevent similar confusion by future readers. I still think that the au-*
*thors could provide additional citations of previous works on the topic.*

Thank you very much.

Main

- 1. *The TOA imbalance is high in some of these supplemental simulations.*
*You should mention the TOA imbalances in the text.*

We have added the TOA imbalance in the caption of the suppl. Fig-
ure 4. The time mean and global averaged TOA imbalances for the
simulations are 0.43 Wm^{-2} for the “6000BCE”, 0.49 Wm^{-2} for the
“6000BCE+20ppm” and 0.38 Wm^{-2} for the “1850CEghg6000BCE”.

Positive values indicate a gain of energy for the atmosphere.

- 2. *You should mention the calendar you used and why the associated bi-*
*ases are likely of secondary importance. It still could be a significant*

*issue for figure 3 c) and d).*

We have used the Proleptic Gregorian calendar. We have not corrected
the seasonal values. Instead we tested the effect of choosing JAS and
JJAS seasonal means values. The different season length did not have
an influence on our interpretation of the results. We have added the fol-
lowing in the Methods section: “The climate model uses the Proleptic
Gregorian calendar.”

3. *So, crops are prescribed but everything else is dynamic? Out of curios-*
*ity, do you get a green Sahara? Does this influence dust? Generally, I*
*think that the contribution from aerosols might be worth expanding.*

Yes, crops are prescribed but everything else is dynamic. This “green“
Sahara is addressed by a new publication which is already submitted.

4. *I cannot read the yellow text in the figures.*

We have changed the colour to enhance the readability.

5. *I am not sure that you need figure 1 b).*

We have removed the figure.

6. *The comparisons with FAMOUS and TraCE simulations is interesting.*

*I think this comparison is worth mentioning in the main text and in-*
*cluding in the supplemental material.*

We have added the additional analyses for the FAMOUS and TRACE
simulations in the supplement of the manuscript and the corresponding
results are now discussed in the main manuscript.

7. *Old text for figure 3 is still in revised manuscript.*

Sorry, it is not clear to us what you mean by old text.

8. *How will people be able to access the model data?*

We will put the corresponding data into a publicly available data
archive.

Response to the reviewer #3:

*This is the review of the revision of "Global temperature modes shed light on*
*the Holocene temperature conundrum" by Bader et al.*

*The authors used MPI-ESM for both transient simulation of past 8000*
*years and three sensitivity experiments to investigate the Holocene temper-*
*ature conundrum. The major finding is that a global warming mode and a*
*cooling mode are the first two EOFs of annual temperature variability from*
*the transient simulation, and the authors attribute the warming mode to the*
*20ppm CO2 increase (and latitudinal trend of annual mean insolation) while*
*the cooling trend to orbitally forced changes in the Arctic sea-ice.*

*Both the transient and sensitivity simulations were very well designed and*
*I commend the authors for completing these long simulations.*

*But I don't think the authors can support their conclusion based on the*
*evidence they provided.*

*My overall criticism for this paper is that there is too much inconsistency*
*between the transient simulation and the two reconstructions that I can't see*
*how the authors can relate the findings from the transient simulation to the*
*two reconstructions.*

Major comments:

1. *In Figure 1, the authors show that the modeled global mean temperature*
*failed to produce the global cooling from the Marcott reconstruction. In*
*order to produce the cooling trend, the authors had to follow what's*
*been done in ref.5 and use summer SST instead of annual SST in the*
*global temperature stack. But as pointed in ref.5, the seasonally biased*
*stack "does not improve model-data inconsistency significantly across*
*individual sites. The spatial correlation of the temperature trends across*
*the 73 sites of Marcott stack between the data, and the ensemble model*
*temperature for the seasonally biased stack (0.01) is not significantly*
*different from that for the annual temperatures (0.16)". So I assume*
*that the seasonal biased simulation artificial mean stack from MPI-ESM*
*wont correct the model-data inconsistency significantly across individual*
*sites.*

*In reply to reviewer 1, the authors acknowledge that "The warming*
*(PC1) and cooling mode (PC2) in our simulation individually show a*
*relative weak relationship to the Marsicek reconstruction."*

*Therefore, the transient simulation fails to catch the major character-*
*istics of either reconstruction, which makes it not credible to relate*
*the warming mode and cooling mode in the transient simulation to the*

*mechanisms behind the trend in the proxy reconstructions.*

Thank you very much for your comments. We agree that differences
between the cooling mode in our simulation and the Marcott recon-
struction still exist. But the Marcott reconstruction might not solely
be an artefact. We can confirm that it is likely that some proxies used
for the reconstructions are biased towards the summer season, but it
does not rule out that at least part of the cooling shown by the Mar-
cott reconstruction has a physical cause. Our model simulation and the
physical explanation supports the existence of a cooling mode. Please
also see the comment to reviewer1.

2. *Another criticism is related to the authors' attribution of warming mode*
*to the 20ppm CO2 increase (and latitudinal trend of annual mean in-*
*solation). This is not correct and not consistent with the results of*
*sensitivity simulations shown in supplemental Figure 4.*

*In fact, the warming mode of PC1 with maximum warming in Tropical*
*Africa and India in Figure 2a is mostly due to the changing orbital pa-*
*rameters and the associated monsoon changes, which the authors have*
*already demonstrated clearly in their sensitivity experiments in Figure*
*S4. Figure S4a shows the pattern of the warming from greenhouse gases*
*with the classic pattern of polar amplification, but doesn't show hardly*

*any warming in the aforementioned monsoon region in Tropical Africa*
*and India in the warming mode in Figure 2a. On the other hand, the*
*warming due to changing the orbital parameters in Figure S4b clearly*
*matches the maximum warming in Tropical Africa and India found in*
*the warming mode of PC1 in Figure 2a, which is in the monsoon re-*
*gions that is consistent with the monsoon changes from changing orbital*
*parameters. Therefore, the sensitivity experiments clearly demonstrate*
*that the warming mode in PC1 is due to changing the orbital parame-*
*ters, not due to 20ppm CO2 increase (and latitudinal trend of annual*
*mean insolation).*

*In summary, both the warming mode and the cooling mode are asso-*
*ciated with the changes in orbital parameters, and EOF analysis of*
*transient simulation shows that the first two leading mode of variabil-*
*ity in the transient simulation are both due to the changes of orbital*
*parameters.*

We agree that also changes in the orbital parameters influence the
warming mode. We tried to make that more clear in the revised ver-
sion. What we want to say is the following: The changes in the orbital
parameters induce not only changes in the seasonal cycle, but also
changes in the annual mean. On the one hand you have the effect

of seasonal changes. This – as you mentioned – is clearly visible in
the monsoon regions. E.g. during the mid-Holocene the boreal sum-
mer insolation was enhanced over the West African Monsoon region
leading to more precipitation and therefore more cooling at the sur-
face. But there is a second effect of the orbital changes which have
an impact on the latitudinal trend of the mean insolation. Over time
the annual mean insolation increases in the tropics. This insolation
increase leads to a tropical warming. The tropical warming leads to
more convection over the tropical oceans thereby enhancing the mid- to
upper-level tropospheric temperatures due to latent heat release. This
upper-tropospheric warming is distributed by atmospheric waves. The
upper-tropospheric warming enhances the stability in the tropics and
therefore leads in particular over the West African monsoon region to
a rainfall reduction by suppressing the convection (please see Figure S7
in Park, Bader, Matei: Anthropogenic Mediterranean warming essen-
tial driver for present and future Sahel rainfall; Nature Climate Change
volume 6, pages 941945 (2016)). This leads therefore to a strong warm-
ing in the monsoon regions. The warming in the tropics is amplified by
the increase in greenhouse gases. In contrast, in the extra-tropics the
annual mean insolation is reduced over time leading to a reduction in
temperatures. This reduction in temperature is partly compensated by

the warming induced by the increase in greenhouse gases. Therefore the
warming mode is influenced by the increase in greenhouse gases and
orbital changes - meaning changes in the latitudinal trend of annual
mean insolation.

Minor comments:

1. *change "effect" to "affect" in L104*

corrected

2. *change "D. Shakun, J." to "Shakun, J. D."*

changed

3. *Add a note on whether the change of the ice sheets topography is in-*
*cluded in the transient simulation between 8000 years ago and 6000*
*years ago.*

We have added the following sentence in the Methods section: "Changes
of the ice sheets topography is not included in the transient simulation."

Reviewers' comments, third round:

Reviewer #3 (Remarks to the Author):

This is my second review of Bader et al "Global temperature modes shed light on the Holocene temperature conundrum". I'm not satisfied with the authors' reply to my comment on the climatic forcing for EOF1. The authors still insist that the EOF1 is from the 20ppm CO2 increase. That's totally WRONG and contradictive to their own sensitivity experiment in suppl. Fig4a with only 20ppm CO2 increase.

I see that both Reviewer 1 and I share the same concern that the EOF1 doesn't exhibit the polar amplification pattern of warming from CO2 forcing. In my previous review, I pointed out that the results from the two sensitivity experiments (Suppl. Fig4) are the exact evidences to prove that the EOF1 is from the response to orbital insolation changes, not from CO2 changes.

Let me illustrate below why EOF1 is not due to CO2 forcing: if you compare the pattern of EOF1 with that from 20ppm-CO2 experiment (Suppl. Fig4a)

EOF1: Major warming occurs in Tropical Africa, India, Alaska and equatorial Pacific

20ppm-CO2 experiment: No warming occurs in Tropical Africa, India, equatorial Pacific

Therefore, Suppl. Fig4a shows CO2 forcing can't produce any warming over three of the four major warming regions from EOF1--Tropical Africa, India and equatorial Pacific.

Instead, orbital experiments in Suppl. Fig4b shows the orbital insolation change can produce the warming in three of the four major warming regions in EOF1: Tropical Africa, India, equatorial Pacific

In a nutshell, orbital insolation changes are responsible for both EOF1 and EOF2.

In fact, this is exactly what is shown in Suppl. Fig4b of the orbital insolation experiment with EOF1-like warming in the Tropical African/India/Equatorial Pacific and EOF2-like cooling in the Arctic region, particularly on the Eurasian side.

I also want to point out that orbital insolation-induced temperature change pattern with cooling in Arctic and warming in Tropical African/India/Equatorial Pacific is very similar to the previous results in citation 5 (their FigS1d) and Otto-Bliesner et al 2006 (their Fig2). So it's been very well documented in the literature that orbital insolation changes are responsible for what's shown in both EOF1 and EOF2.

Additional comments:

1. L85, add "with the acceleration of climatic forcing by a factor of 10" after "albeit at coarse resolution"
2. L206, sentence is not complete.

Reviewer #4 (Remarks to the Author):

I am ok with the paper for publication now

Reviewer #5 (Remarks to the Author):

I was asked to come in as an additional reviewer and to read through the manuscript, supplementary material, the reviews, the authors' response, and the 3rd reviewers reply to the

response. I have now done that, and will focus my review on crux of the matter, without a broader review of the manuscript.

It's a strong manuscript with interesting results. The key point of contention is on the interpretation of PC1. In the manuscript the authors associate PC1, the warming mode, with the greenhouse effect, largely driven by increasing CO₂ through the Holocene, contrasting with the cooling mode, associated with orbital forcing driving cooling in the high latitudes. However, as reviewer 3 points out, there is little evidence that PC1 is primarily associated with CO₂. Instead, the sensitivity analysis in Supp. Fig 4 suggests that the high loading regions of PC1 are more closely associated with orbital forcing, and matches poorly with CO₂ forcing. The authors do note in their reply that the mode is a balance of these two forcings, which is of course true, PC1 will integrate the effects of both forcings. However, it seems that orbital is the dominant effect driving the warming seen in PC1, not greenhouse gas forcing. Indeed, I suspect that if the simulation was run without changes in CO₂, you'd see similar results in PC1. Therefore, the characterization of PC1 as a greenhouse gas driven effect is hard to justify.

That said, viewing the conundrum through the lens of two, primarily orbital, modes, one with cooling driven by orbital-induced ice-albedo feedbacks primarily in the Arctic, and another with orbital precipitation and cloud albedo interactions at lower latitudes is still an interesting and useful contribution. As far as novelty, others have noted that the warming through the Holocene isn't uniform (cooling in the high-latitudes is frequently simulated) and the PC1 warming patterns have been observed before. That said, the presentation here is a useful expansion on that theme. In terms of changing the interpretation to better match the forcings, it is a little less satisfying than a CO₂-driven mode, as CO₂-driven warming is thought to be a large component of the conundrum.

1 Response to the reviewers of the manuscript

2 “Global temperature modes shed light on the

3 Holocene temperature conundrum”

4 Jürgen Bader, Johann Jungclaus, Natalie Krivova,

Stephan Lorenz, Amanda Maycock, Thomas Raddatz,

Hauke Schmidt, Matthew Toohey, Chi-Ju Wu, Martin Claussen

January 8, 2020

Reviewer #3:

This is my second review of Bader et al ”Global temperature modes shed

light on the Holocene temperature conundrum”. I’m not satisfied with the

authors’ reply to my comment on the climatic forcing for EOF1. The authors

still insist that the EOF1 is from the 20ppm CO2 increase. That’s totally

WRONG and contradictive to their own sensitivity experiment in suppl.

Fig4a with only 20ppm CO2 increase.

I see that both Reviewer 1 and I share the same concern that the EOF1
doesn't exhibit the polar amplification pattern of warming from CO2 forcing.
In my previous review, I pointed out that the results from the two sensitivity
experiments (Suppl. Fig4) are the exact evidences to prove that the EOF1
is from the response to orbital insolation changes, not from CO2 changes.
Let me illustrate below why EOF1 is not due to CO2 forcing: if you compare
the pattern of EOF1 with that from 20ppm-CO2 experiment (Suppl. Fig4a)
EOF1: Major warming occurs in Tropical Africa, India, Alaska and equato-
rial Pacific 20ppm-CO2 experiment: No warming occurs in Tropical Africa,
India, equatorial Pacific
Therefore, Suppl. Fig4a shows CO2 forcing can't produce any warming over
three of the four major warming regions from EOF1–Tropical Africa, India
and equatorial Pacific.
Instead, orbital experiments in Suppl. Fig4b shows the orbital insolation
change can produce the warming in three of the four major warming regions
in EOF1: Tropical Africa, India, equatorial Pacific
In a nutshell, orbital insolation changes are responsible for both EOF1 and
EOF2.
In fact, this is exactly what is shown in Suppl. Fig4b of the orbital insolation
experiment with EOF1-like warming in the Tropical African/India/Equatorial
Pacific and EOF2-like cooling in the Arctic region, particularly on the Eurasian

side.

I also want to point out that orbital insolation-induced temperature change
pattern with cooling in Arctic and warming in Tropical African/India/Equatorial
Pacific is very similar to the previous results in citation 5 (their FigS1d) and
Otto-Bliesner et al 2006 (their Fig2). So it's been very well documented in
the literature that orbital insolation changes are responsible for what's shown
in both EOF1 and EOF2.

Additional comments:

- 1. L85, add "with the acceleration of climatic forcing by a factor of 10" after
"albeit at coarse resolution"
- 2. L206, sentence is not complete.

Reviewer #5:

I was asked to come in as an additional reviewer and to read through the
manuscript, supplementary material, the reviews, the authors' response, and
the 3rd reviewers reply to the response. I have now done that, and will focus
my review on crux of the matter, without a broader review of the manuscript.
It's a strong manuscript with interesting results. The key point of contention
is on the interpretation of PC1. In the manuscript the authors associate PC1,

the warming mode, with the greenhouse effect, largely driven by increasing
CO₂ through the Holocene, contrasting with the cooling mode, associated
with orbital forcing driving cooling in the high latitudes. However, as re-
viewer 3 points out, there is little evidence that PC1 is primarily associated
with CO₂. Instead, the sensitivity analysis in Supp. Fig 4 suggests that the
high loading regions of PC1 are more closely associated with orbital forcing,
and matches poorly with CO₂ forcing. The authors do note in their reply
that the mode is a balance of these two forcings, which is of course true, PC1
will integrate the effects of both forcings. However, it seems that orbital is
the dominant effect driving the warming seen in PC1, not greenhouse gas
forcing. Indeed, I suspect that if the simulation was run without changes
in CO₂, you'd see similar results in PC1. Therefore, the characterization of
PC1 as a greenhouse gas driven effect is hard to justify.

That said, viewing the conundrum through the lens of two, primarily orbital,
modes, one with cooling driven by orbital-induced ice-albedo feedbacks pri-
marily in the Arctic, and another with orbital precipitation and cloud albedo
interactions at lower latitudes is still an interesting and useful contribution.
As far as novelty, others have noted that the warming through the Holocene
isn't uniform (cooling in the high-latitudes is frequently simulated) and the
PC1 warming patterns have been observed before. That said, the presen-
tation here is a useful expansion on that theme. In terms of changing the

interpretation to better match the forcings, it is a little less satisfying than
a CO₂-driven mode, as CO₂-driven warming is thought to be a large com-
ponent of the conundrum.

Response to the reviewers #3 and #5 :

We totally agree with the reviewers that there is a substantial effect of the
orbital forcing in the spatial pattern of the warming mode.

The reviewers mainly criticise that the interpretation of PC1 as green-
house gas dominated is not justified enough and that this mode is likely
dominated by orbital forcing. Apparently the text is not written clearly
enough so that this misunderstanding occurs. The reviewers seem to talk
about the spatial pattern of the warming mode. E.g. one reviewer refers
to suppl. Fig4 that shows spatial patterns. We totally agree that there is
a substantial effect of both greenhouse-gas and orbital forcing in the spatial
pattern of the warming mode. Significant imprints of the orbital forcing are
visible in the spatial warming pattern. For example a substantial anomalous
warming in the tropics and a clear anomalous cooling in the polar regions
is caused by the orbital forcing (please see suppl. Fig4). We want to say
that the time evolution - not the spatial pattern - seems to be dominated by
the time evolution of the greenhouse gas concentrations. The first mode of

the EOF analysis mainly picks up the covariability that is associated with
the temporal change in the greenhouse gas forcing. This is indicated by the
strong correlation between the time evolution of the warming mode (PC1)
and the greenhouse effect (Fig3). This does not exclude effects of the orbital
forcing on the spatial pattern of the warming mode. The spatial pattern of
the warming mode incorporates the effect of the greenhouse gas forcing and
that part of the effect of the orbital forcing that covaries with the temporal
change in the greenhouse gas forcing. To summarise: the spatial warming
pattern clearly shows the effects of orbital and greenhouse gas forcing. The
time evolution of the warming mode seems to be dominated by the temporal
change in greenhouse gas concentrations.

We have changed the text to hopefully avoid the misunderstanding. In ad-
dition, we modified the caption of Figure 3 – now Figure 4.

1. L85, add "with the acceleration of climatic forcing by a factor of 10" after
"albeit at coarse resolution"

done

2. L206, sentence is not complete.

The sentence is now complete.

Reviewers' comments, fourth round:

Reviewer #5 (Remarks to the Author):

Again, I'll get right to the major issue, the interpretation of PC1 as a CO₂-driven Holocene warming mode; despite the evidence that this mode is strongly controlled by insolation. In their revision, the authors argue that, yes there is a significant influence of orbital variability on the spatial component of PC1, but the temporal component of PC1 is driven by changes in CO₂ and the greenhouse effect. The primary evidence for the second half of this argument is that the PC1 timeseries very closely resembles changes in CO₂ and the energy balance changes association with GHG in the model.

I have two problems with this argument.

1. In EOF analysis, you can't neatly separate the causes of the spatial pattern from the causes of the temporal pattern; the temporal pattern is literally the weighted average of global temperatures, with the spatial loading pattern is the weights. So if orbital changes are known or shown to drive changes in certain regions, which is largely where the loadings are in EOF1, then the temporal changes in PC1 (which is largely just an average of temperatures in those same regions) must also be caused by orbital changes. It's like arguing that, yes, A causes changes in B, but the changes we see in B are not caused by A, instead they're caused by C, because the patterns are similar.

2. If the best evidence that the PC1 time-evolution is driven by changes in CO₂ is that the temporal patterns of the two phenomena are very similar, I'd like to propose an alternative hypothesis. The time evolution of Northern Hemisphere JJA insolation in the Holocene (and Global mean annual insolation) also strongly resemble PC1 (see attached figure). I think there are two reasons why this is a far more plausible explanation for temporal of PC1. First, the primary changes in EOF1 occur in the monsoon regions of North Africa and India, which are very sensitive to changes in summer insolation. Second, the W/m² change in summer NH forcing is about 20 times larger than the change in greenhouse warming over the same interval (28 vs 1.4 W/m²), meaning that it's a likely to be a much larger perturbation to global climate.

As I mentioned in my last review, I think the path forward is to describe both EOFs (and PCs) as primarily orbitally driven modes; as I think there's not enough evidence that either mode is driven by CO₂. It's still an interesting contribution as such; and is on much stronger footing in terms of interpreting the results.

Response to the reviewers of the manuscript “Global temperature modes shed light on the Holocene temperature conundrum”

Jürgen Bader, Johann Jungclaus, Natalie Krivova,
Stephan Lorenz, Amanda Maycock, Thomas Raddatz,
Hauke Schmidt, Matthew Toohey, Chi-Ju Wu, Martin Claussen

March 17, 2020

Reviewer #5

Again, I'll get right to the major issue, the interpretation of PC1 as a CO₂-driven Holocene warming mode; despite the evidence that this mode is strongly controlled by insolation. In their revision, the authors argue that, yes there is a significant influence of orbital variability on the spatial component of PC1, but the temporal component of PC1 is driven by changes in CO₂ and the greenhouse effect. The primary evidence for the second half of this argument is that the PC1 timeseries very closely resembles changes in CO₂ and the energy balance changes association with GHG in the model.

I have two problems with this argument.

1. In EOF analysis, you can't neatly separate the causes of the spatial pattern from the causes of the temporal pattern; the temporal pattern is literally the weighted average of global temperatures, with the spatial loading pattern is the weights. So if orbital changes are known or shown to drive changes in certain regions, which is largely where the loadings are in EOF1, then the temporal changes in PC1 (which is largely just an average of temperatures in those same regions) must also be caused by orbital changes. It's like arguing that, yes, A causes changes in B, but the changes we see in B are not caused by A, instead they're caused by C, because the patterns are similar.

2. If the best evidence that the PC1 time-evolution is driven by changes in CO₂ is that the temporal patterns of the two phenomena are very similar, I'd like to propose an alternative hypothesis. The time evolution of Northern Hemisphere JJA insolation in the Holocene (and Global mean annual insolation) also strongly resemble PC1 (see attached figure). I think there are two reasons why this is a far more plausible explanation for temporal of PC1. First, the primary changes in EOF1 occur in the monsoon regions of North Africa and India, which are very sensitive to changes in summer insolation. Second, the W/m² change in summer NH forcing is about 20 times larger than the change in greenhouse warming over the same interval (28 vs 1.4 W/m²), meaning that it's a likely to be a much larger perturbation to global climate.

As I mentioned in my last review, I think the path forward is to describe both EOFs (and PCs) as primarily orbitally driven modes; as I think there's not enough evidence that either mode is driven by CO₂. It's still an interesting contribution as such; and is on much stronger footing in terms of interpreting the results.

Response to the Reviewer#5

We will give evidence that the temporal evolution of PC1 is very likely dominated by the greenhouse effect and not by orbital forcing. This conclusion/evidence is based on the current literature/understanding and on our analysis.

Evidence from the literature

As Reviewer#5 pointed out there is a clear connection between the simulated global mean temperature and PC1 (the temporal evolution of the spatial pattern explaining the largest amount of variability): " ... the temporal pattern is literally the weighted average of global temperatures, with the spatial loading pattern is the weights." Actually, the temporal PC1 weighted by the spatially averaged EOF (the corresponding spatial pattern) represents the largest contribution to the simulated annual global mean temperature. Several studies have shown that the simulated global annual mean warming during the Holocene is mainly caused by an increase in greenhouse gas concentrations and by the retreat of ice sheets (which are not considered in

our simulation because the simulation starts at the beginning of the mid-Holocene). E.g. Liu et al (2014) write: "Here, we show that climate models simulate a robust global annual mean warming in the Holocene, mainly in response to rising CO₂ and the retreat of ice sheets." The results by Liu et al. 2014 are based on simulations with three different models and hence indicate the robustness of this result. Therefore, the assertion of Reviewer#5 that **PC1 is mainly orbitally driven cannot be reconciled with the current state of knowledge** that the simulated global mean temperature is mainly forced by changes in greenhouse gases.

Theoretical considerations and experimental evidence

Reviewer#5 argues that the orbitally driven changes are larger than the CO₂ forcing changes (28 vs 1.4 Wm⁻²). This is valid, only, if changes in mid-latitude, summer insolation are considered. The discussion in our paper, however, focuses on annual mean temperature changes, but not on seasonal nor on regional changes. If we consider the annual global mean insolation changes during the mid-Holocene, then we find that the low-frequency annual global mean **orbital forcing changes are marginal** (Figure 1a in this response). The low-frequency changes of the annual global mean solar insolation range between -0.05 to 0.03 Wm⁻². Therefore the orbital forcing changes are about one to two orders of magnitude smaller than the changes in the greenhouse effect (1.5 Wm⁻²; see Figure 1b in this response). Hence, no substantial simulated annual global mean warming due to changes in the orbital forcing is expected. This is tested in sensitivity experiments (see further responses). The dominant changes in the greenhouse gas forcing strength – compared to the changes in the orbital forcing strength – might explain the good agreement between the time evolution of PC1 and the greenhouse effect shown in Figure 1b of this response. Therefore, our analysis – based on a model not used in the study by Liu2014 – clearly confirms the importance of the greenhouse effect for the warming during the mid-Holocene.

To find further evidence for the dominant role of the greenhouse gas forcing changes we analysed additional sensitivity experiments already introduced in the supplementary material. In addition to the transient simulation, three other time-slice experiments are performed with the MPI-ESM to better disentangle the effect of the CO₂ increase and the influence of the change in the orbital parameters from the beginning (6000 BCE) to the end (1850 CE) of the transient simulation. Each sensitivity experiment is run for 1000 years. One Holocene sensitivity experiment is performed with the orbital forcing

Figure 1: **Low-frequency change in the annual global mean incoming solar radiation is marginal.** (a) Low-pass filtered anomaly of the annual global mean incoming solar radiation at the top of the atmosphere [Wm^{-2}]. (b) The grey curve shows the low-pass filtered greenhouse effect and the red curve the low-pass filtered warming mode (PC1). The greenhouse gas effect is defined as the difference between the upward surface thermal radiation and the outgoing longwave radiation at the top of the atmosphere [Wm^{-2}]

and the greenhouse gas concentration at 6000 BCE. The simulation is called “6000BCE”. A second simulation is done with the same orbital forcing, but the atmospheric CO_2 concentration is enhanced by 20 *ppm*. This enhancement corresponds approximately to the difference between the CO_2 concentration at the beginning (6000 BCE) and the end (1850 CE) of the transient simulation. We refer to this simulation as “6000BCE+20ppm”. The simulations “6000BCE” and “6000BCE+20ppm” differ only in the CO_2 concentrations. A third Holocene simulation is performed with the orbital forcing at 1850 CE but with the greenhouse gas concentrations (CO_2 , CH_4 , N_2O) at 6000 BCE. The simulation is called “1850CEghg6000BCE”.

Suppl. Figure 4a shows the near-surface annual mean temperature difference between the simulations “6000BCE+20ppm” and “6000BCE”. Therefore, it shows the temperature change caused by a 20 *ppm* CO_2 increase. Suppl. Figure 4b shows the near-surface temperature difference between the simulations “1850CEghg6000BCE” and “6000BCE”. The temperature change in the annual mean temperature is caused by changing the orbital parameters at 6000 BCE to 1850 CE values.

We have computed the simulated global annual mean temperature change induced individually by orbital forcing changes or CO_2 forcing changes. The simulated annual global mean temperature change caused by a 20 *ppm* CO_2 increase is about **0.3** K. In contrast, the simulated annual global mean temperature change induced by changes in the orbital forcing is about **0** K. Therefore, these sensitivity experiments indicate that the simulated annual global mean temperature change is dominated by the CO_2 forcing change and not by the changes in the orbital forcing. In addition, the sensitivity experiments confirm the results by Liu2014 and our theoretical considerations. The more or less absence of any simulated annual global mean warming or cooling in response to orbital changes confirms our theoretical considerations. As we have argued before, the change in the annual global mean temperature is associated with the annual global mean insolation changes and not with seasonal and regional changes. Therefore, we would not expect any larger changes of the annual global mean temperature during the mid-Holocene induced by orbital forcing changes, because the annual global mean orbital forcing change is marginal.

Conclusions

Our conclusion that PC1, which represents the largest contribution to the simulated annual global mean temperature increase, is dominated by changes in greenhouse gas forcing is:

1. in agreement with all other studies we are aware of
2. indicated by the high correlation between the warming mode (PC1) and the greenhouse effect
3. because the annual global mean orbital forcing changes are marginal and about one to two orders of magnitude smaller than the changes in the greenhouse effect
4. supported by additional sensitivity experiments showing that the warming of the simulated global annual mean temperature induced by a 20 *ppm* CO_2 increase is about 0.3 K. In contrast, there is no substantial global annual mean temperature increase simulated due to changes in the orbital forcing

Reference

Liu, Zhengyu, Jiang Zhu, Yair Rosenthal, Xu Zhang, Bette L. Otto-Bliesner, Axel Timmermann, Robin S. Smith, Gerrit Lohmann, Weipeng Zheng, and Oliver Elison Timm. The holocene temperature conundrum. *Proceedings of the National Academy of Sciences*, 111(34):E3501E3505, 2014. doi: 10.1073/pnas.1407229111.

Changes to the article

We have added the line (see line 150): "The temporal PC1 weighted by the corresponding spatially averaged EOF represents the largest contribution to the simulated annual global mean temperature."

We have added the following paragraph (lines 156-176):

"To find further evidence for the dominant role of the greenhouse gas forcing changes we analysed additional sensitivity experiments (for details see sensitivity experiments in the supplement). Three time-slice experiments are performed with the MPI-ESM to better disentangle the effect of the CO_2 increase and the influence of the change in the orbital parameters from the beginning (6000 BCE) to the end (1850 CE) of the transient simulation. One Holocene sensitivity experiment is performed with the orbital forcing

and the greenhouse gas concentration at 6000 BCE. A second simulation is done with the same orbital forcing, but the atmospheric CO_2 concentration is enhanced by 20 *ppm*. This enhancement corresponds approximately to the difference between the CO_2 concentration at the beginning (6000 BCE) and the end (1850 CE) of the transient simulation. A third Holocene simulation is performed with the orbital forcing at 1850 CE but with the greenhouse gas concentrations (CO_2 , CH_4 , N_2O) at 6000 BCE. By calculating the differences, we can determine the simulated global mean temperature change due to the CO_2 increase and due to the change in orbital parameters. The simulated annual global mean temperature change caused by a 20 *ppm* CO_2 increase is about 0.3 K. In contrast, the simulated annual global mean temperature change induced by changes in the orbital forcing is about 0 K. Therefore, these sensitivity experiments indicate that the simulated annual global mean temperature change is dominated by the CO_2 forcing change.”

Reviewers' comments, fifth round:

Reviewer #5 (Remarks to the Author):

I appreciate the authors' response, and especially the new simulations which I think shed additional light on the issue at hand. However, I continue to disagree that the evidence points to increases in GHGs as the driver of PC1. In their rebuttal, the authors conclude that PC1 "is dominated by changes in greenhouse gas forcing" for four reasons:

1. (it is) in agreement with all other studies we are aware of
2. indicated by the high correlation between the warming mode (PC1) and the greenhouse effect
3. because the annual global mean orbital forcing changes are marginal and about one to two orders of magnitude smaller than the changes in the greenhouse effect
4. supported by additional sensitivity experiments showing that the warming of the simulated global annual mean temperature induced by a 20 ppm CO₂ increase is about 0.3 K. In contrast, there is no substantial global annual mean temperature increase simulated due to changes in the orbital forcing

I think that there are a couple misunderstandings that are leading to our continued disagreement. First - I agree that increases in GHG concentrations are driving the trend towards increased temperature in the simulation, and other studies have found the same thing. However, just because GHG increases are driving the first-order trend in GMST, does not necessarily mean that GHG increases are causing the variability in PC1. That CO₂ drives simulated Holocene warming is well established; but the novelty of this article is that it claims to detect competing modes in the simulation, the first reflecting GHG forcing. This distinction is important. The EOF analyses conducted here are designed to extract the primary modes of spatiotemporal temperature variability, which does not have to respond to the same forcing that drives the long term trend. It's not a trend analysis, it's an analysis looking for the leading modes of variability. That the cause of a long term trend may not be the driver of the leading mode of variability can be easily demonstrated numerically in a simple dataset.

The new sensitivity experiments are an excellent addition to the study, and I appreciate the authors effort conducting them. They support my argument that PC1 is not driven by CO₂ increases. The spatial pattern of PC1 (Fig 2a) resembles the orbital only simulation (Supp Fig 4b), and does not at all resemble the CO₂ increase experiment (Supp Fig 4a). This should pretty clear indicate that the spatial pattern associated with temperature changes in the orbital only experiment is the same as in PC1. Why should we expect that pattern to reflect temperature response to orbital changes in the sensitivity experiment, but not in the full simulation?

I wonder if part of the misunderstanding is confusion between the changes driven by the greenhouse effect, and those driven by increased GHG concentrations. Supp Fig 5 shows large changes in the greenhouse effect in the orbital only simulation, but critically in the opposite direction than in PC1. Specifically, the monsoon regions (especially N Africa and India) show a much larger "greenhouse effect" at 6 ka than the modern. Whereas in PC1, those regions are responsible for driving the increase in PC1 through the Holocene. This is all consistent with my hypothesis that PC1 is largely driven by a reduction in cloud cover over those monsoon systems, which would increase surface temperatures (by lowering albedo), and reduce the greenhouse effect by reducing cloud cover and water vapor in those regions. These systems are known to respond to insolation (as demonstrated in Supp Fig 5), and not CO₂ changes.

Ultimately it comes down to this. If the authors would like to argue that simulated Holocene GMST trends were driven by CO₂ increases, that's fine, I agree, but it's well established and not novel. In the manuscript, the authors claim to detect a CO₂ warming mode in the simulation using EOF analysis, but the spatial pattern of that mode is both inconsistent with CO₂ increases (which should resemble supp Fig 4a), and very consistent with insolation-driven changes in temperature patterns (like supp Fig 4b), which don't cause a net change in GMST, but are large changes and are a leading mode of variability.

Coming back to their four reasons in support of their hypothesis, I'll respond to each in turn:

1. (it is) in agreement with all other studies we are aware of

That CO₂ drives the leading mode of variability (not the cause of the net warming) is not established (indeed, it's the proposed novelty of this paper).

2. indicated by the high correlation between the warming mode (PC1) and the greenhouse effect

The correlation with insolation is just as good.

3. because the annual global mean orbital forcing changes are marginal and about one to two orders of magnitude smaller than the changes in the greenhouse effect

The large and asymmetric changes in seasonal insolation can drive large changes in the spatial pattern of temperature, making it the leading mode of variability, while averaging out to small annual changes (see supp fig 4b).

4. supported by additional sensitivity experiments showing that the warming of the simulated global annual mean temperature induced by a 20 ppm CO₂ increase is about 0.3 K. In contrast, there is no substantial global annual mean temperature increase simulated due to changes in the orbital forcing

These sensitivity experiments clearly show that the spatial pattern associated with GHG increase does not match PC1; whereas the spatial pattern associated with insolation change matches clearly.

Response to the reviewers of the manuscript “Global temperature modes shed light on the Holocene temperature conundrum”

Jürgen Bader, Johann Jungclaus, Natalie Krivova,
Stephan Lorenz, Amanda Maycock, Thomas Raddatz,
Hauke Schmidt, Matthew Toohey, Chi-Ju Wu, Martin Claussen

July 2, 2020

There are continuous misunderstandings and conflicts concerning the interpretation of the warming mode that we identified in our simulations, namely whether it is caused by rising CO₂ and/or by orbital insolation changes. But this is only a side aspect of our paper and does not touch our main finding that the occurrence of a warming and a cooling mode can solve the Holocene temperature conundrum. Therefore, we can omit the conflicting interpretation of the warming mode from the paper without affecting its core scientific results.

Following this suggestion, in the resubmitted paper we now abstain from any interpretation of the warming mode. To this end, we removed all text where the CO₂-rise was claimed to drive the warming mode. Consequently, we also removed from the supplement all the material that was meant to support our interpretation of the warming mode.

As a final remark we may add that most of the text and material now removed was not part of the original submission, but entered the manuscript in reaction to the reviewers concern about our interpretation of the origin of the warming mode.

The following comments refer to the manuscript version of March. With our answers we want to clear up the misunderstandings and conflicts that have arisen.

Reviewer #5

I appreciate the authors response, and especially the new simulations which I think shed additional light on the issue at hand. However, I continue to disagree that the evidence points to increases in GHGs as the driver of PC1. In their rebuttal, the authors conclude that PC1 is dominated by changes in greenhouse gas forcing for four reasons:

1. (it is) in agreement with all other studies we are aware of
2. indicated by the high correlation between the warming mode (PC1) and the greenhouse effect
3. because the annual global mean orbital forcing changes are marginal and about one to two orders of magnitude smaller than the changes in the greenhouse effect
4. supported by additional sensitivity experiments showing that the warming of the simulated global annual mean temperature induced by a 20 ppm CO₂ increase is about 0.3 K. In contrast, there is no substantial global annual mean temperature increase simulated due to changes in the orbital forcing

I think that there are a couple misunderstandings that are leading to our continued disagreement. First - I agree that increases in GHG concentrations are driving the trend towards increased temperature in the simulation, and other studies have found the same thing. However, just because GHG increases are driving the first-order trend in GMST, does not necessarily mean that GHG increases are causing the variability in PC1. That CO₂ drives simulated Holocene warming is well established; but the novelty of this article is that it claims to detect competing modes in the simulation, the first reflecting GHG forcing. This distinction is important. The EOF analyses conducted here are designed to extract the primary modes of spatiotemporal temperature variability, which does not have to respond to the same forcing that drives the long term trend. Its not a trend analysis, its an analysis looking for the leading modes of variability. That the cause of a long term trend may not be the driver of the leading mode of variability can be easily demonstrated numerically in a simple dataset.

We more or less agree of what the reviewer has written and we think that our results are not in contradiction. Nevertheless, because we also think that there are a couple of misunderstandings, we briefly repeat the key point of

contention and our point of view on the forcing of the warming mode.

What is the key point of contention?

The 'conflict' or the key point of contention is on the interpretation of the warming mode (the first mode of the EOF analysis) – although we think this is only a side aspect of the paper. The reviewer mainly criticises that the interpretation of the warming mode as greenhouse gas dominated is not justified enough and that this mode is likely dominated by orbital forcing. We have the impression that our interpretation/explanation of the drivers of the warming mode has not come across correctly. We assume that we have not made it clear enough how we discriminate between spatial and temporal variations in the interpretation of the warming mode. As we had defined in the manuscript, we use the term 'PC' just for the temporal variability, whereas the reviewer uses 'PC' also for the spatial variability pattern. In the manuscript, we defined to use the term 'PC' for the temporal development whereas the term 'EOF' is used for the spatial pattern. Our impression is that when we are talking about the temporal development/variability this interpretation is also transferred to the spatial variability pattern. E.g. in the previous response to the reviewer we were only talking about the temporal variability and not the spatial variability pattern.

What is our point of view on the forcing of the warming mode?

We agree with the reviewer that there is a substantial effect of the orbital forcing in the spatial pattern of the warming mode (EOF1). Significant imprints of the orbital forcing are visible in the spatial warming pattern (EOF1). For example a substantial anomalous warming in the tropics and a clear anomalous cooling in the polar regions is caused by the orbital forcing (please see e.g. the lines 176 to 197 in the previous/the reviewed version of the manuscript).

We want to say that the time evolution (PC1) - not the spatial pattern (EOF1) - seems to be dominated by the time evolution of the greenhouse gas concentrations. The first mode of the EOF analysis mainly picks up the covariability that is associated with the temporal change in the greenhouse gas forcing. This is indicated by the strong correlation between the time evolution of the warming mode (PC1) and the greenhouse effect (Figure 4 in the current version). This does not exclude effects of the orbital forcing on the spatial pattern of the warming mode. The spatial pattern of the warming mode incorporates the effect of the greenhouse gas forcing and that part of the effect of the orbital forcing that covaries with the temporal change

in the greenhouse gas forcing. To summarise: the spatial warming pattern (EOF1) clearly shows the effects of orbital and greenhouse gas forcing. The time evolution of the warming mode (PC1) seems to be dominated by the temporal change in greenhouse gas concentrations.

Therefore, we agree with the reviewer that there is a substantial imprint of the orbital forcing in the spatial pattern of the warming mode (EOF1). Hence, we think that our interpretation in the manuscript is not in contradiction to the reviewer's view - but it is a bit more differentiated. In particular, we note that there is also a substantial role for the CO₂ forcing. We think that our more differentiated view is important. If the warming mode – the spatial pattern (EOF1) and the time evolution (PC1) – would be interpreted as being mainly insolation driven, that would imply that also the global mean warming is mainly insolation driven. We have shown that the warming and cooling mode (the first two modes) mainly explain the low-frequency global-mean temperature. The cooling mode is not substantially driven by the CO₂ increase. If the first mode (the spatial pattern (EOF1) and the temporal evolution (PC1)) would be mainly orbital driven, this would require that also the global mean temperature increase (the warming) would be orbital driven. This is not supported by our results and would probably cause irritations/debates in the scientific community, because this interpretation would mean that the CO₂-increase is not responsible for the global mean warming. This is also not supported by the literature. The orbital forcing change leads mainly to a spatial redistribution of the energy. This is clearly visible in the spatial pattern (EOF1) of the warming mode. But the orbital changes do not substantially change the annual global mean energy input - please also see previous response to the reviewer. Therefore, from an energetic point of view the orbital forcing changes cannot significantly contribute to the global mean temperature warming – assuming that the climate system to first order responds linearly. To make our interpretation of the warming mode more clear we have constructed an idealised temperature field and performed afterwards an EOF-analysis (please see Figure 1 in this document). The effect of the CO₂ on the temperature is prescribed as a homogeneous spatial field that increases during the first half and is constant during the second half. This development is very characteristic and should be easy to recognise in the time evolution (PC) of the empirical orthogonal function analysis. In addition, the time evolution is somehow also similar to our transient simulation, because most of the CO₂ is added during the first half of the simulation. The effect of the orbital forcing is prescribed as a tripole pattern. Continuous warming in the tropics and cooling in the extratropics. The global mean of this temperature field is about 0 K – this is important to note. We have shown in the additional sensitivity experiments

(please see previous supplementary Figure 4 and the previous response to the reviewer) that the orbital forcing effect on the global mean temperature more or less vanishes. This is a consequence of a global mean energy input change close to zero due to the orbital changes (please see Figure 1 in the previous response to the reviewer). We have added the two temperature fields shown in Figure 1 and performed an empirical orthogonal function analysis. The first mode of the EOF analysis explains more than 98% of the variance. The spatial pattern (EOF1) clearly shows the imprint of the orbital forcing. But there is an important difference to the prescribed orbital temperature field. The prescribed orbital temperature field is positive in the tropics and negative in the extratropics. The spatial pattern of the first mode (EOF1) of the EOF-analysis of the idealised temperature field is just positive. It is not a tripole-like pattern as the prescribed idealised orbital temperature field, but a monopole pattern with stronger loadings in the tropics and weaker loadings in the extratropics. The spatial pattern (EOF1) of the idealised temperature field is therefore more a superposition of the idealised prescribed orbital and CO2 temperature field. The time evolution of the first mode (PC1) is dominated by the time evolution of the idealised CO2 increase.

Because the field mean of the spatial pattern of the warming mode (EOF1) of the transient simulation is positive and does not vanish may be an indication that also the CO2-effect is in the spatial pattern of the warming mode (EOF1). In addition, the pattern correlation between the spatial pattern of the warming mode (EOF1) of the transient simulation and the superposition/sum of the two pattern fields of the sensitivity experiments – shown in the previous supplementary Figure 4 – are higher than the pattern correlation with the individual patterns. The CO2-effect in the spatial pattern of the warming mode (EOF1) is needed so that it has a substantial contribution to the global mean temperature.

The new sensitivity experiments are an excellent addition to the study, and I appreciate the authors effort conducting them. They support my argument that PC1 is not driven by CO2 increases. The spatial pattern of PC1 (Fig 2a) resembles the orbital only simulation (Supp Fig 4b), and does not at all resemble the CO2 increase experiment (Supp Fig 4a). This should pretty clear indicate that the spatial pattern associated with temperature changes in the orbital only experiment is the same as in PC1. Why should we expect that pattern to reflect temperature response to orbital changes in the sensitivity experiment, but not in the full simulation?

There is a misunderstanding here. As mentioned above we agree that in the

spatial pattern of the warming mode (EOF1) a substantial effect of the orbital forcing is seen. As we pointed out at the beginning: "We have the impression that our interpretation/explanation of the drivers of the warming mode has not come across correctly. We assume that we have not made it clear enough how we discriminate between spatial and temporal variations in the interpretation of the warming mode. As we had defined in the manuscript, we use the term 'PC' just for the temporal variability, whereas the reviewer uses 'PC' also for the spatial variability pattern. In the manuscript, we defined to use the term 'PC' for the temporal development whereas the term 'EOF' is used for the spatial pattern. Our impression is that when we are talking about the temporal development/variability this interpretation is also transferred to the spatial variability pattern."

I wonder if part of the misunderstanding is confusion between the changes driven by the greenhouse effect, and those driven by increased GHG concentrations. Supp Fig 5 shows large changes in the greenhouse effect in the orbital only simulation, but critically in the opposite direction than in PC1. Specifically, the monsoon regions (especially N Africa and India) show a much larger greenhouse effect at 6 ka than the modern. Whereas in PC1, those regions are responsible for driving the increase in PC1 through the Holocene. This is all consistent with my hypothesis that PC1 is largely driven by a reduction in cloud cover over those monsoon systems, which would increase surface temperatures (by lowering albedo), and reduce the greenhouse effect by reducing cloud cover and water vapor in those regions. These systems are known to respond to insolation (as demonstrated in Supp Fig 5), and not CO₂ changes.

Yes, there is a difference between "greenhouse effect" and "greenhouse gas effect". In the manuscript the "greenhouse effect" is defined "... as the difference between the upward surface thermal radiation and the outgoing longwave radiation at the top of the atmosphere." Therefore the "greenhouse effect" can include orbital changes, because it is temperature dependent.

Nevertheless, there seems to be a misunderstanding here. We agree that in the spatial pattern of the warming mode (EOF1) a substantial effect of the orbital forcing is included. We have also written this in the previous/reviewed version of the manuscript explicitly: "The strong correlation between the time evolution of the warming mode (PC1) and the greenhouse effect does not exclude effects of the orbital forcing on the spatial pattern of the warming mode (EOF1). The spatial pattern of the warming mode incorporates the effect of the greenhouse gas forcing and that part of the orbital forcing effect that covaries with the temporal change in the greenhouse gas forcing. There-

fore, the spatial pattern (EOF1) of the simulated Holocene warming mode differs substantially from the warming pattern seen in future climate scenarios. For example, in contrast to the projected future changes the Holocene warming mode is most pronounced in the tropical regions and hardly shows any Arctic amplification. This difference in the spatial warming pattern during the Holocene is related to the latitude-dependent trend in annual mean insolation forced by orbital changes. Although the global mean annual insolation does not change substantially, strong regional changes exist. For instance, the annual mean incoming solar radiation in the Arctic decreases, whereas the radiation in the tropics increases (supplementary Figure 1a). As a consequence, the greenhouse effect – because it is temperature dependent – is reinforced in the tropics and almost offset in the Arctic (see also supplementary Figures 4,5). **Moreover, globally, the strongest warming occurs in the West African and Indian monsoon regions, because of a positive feedback associated with the reduction in monsoonal rainfall and an associated decrease in evaporative cooling and total cloud cover.**” Therefore, we think that our point of view is in line with the reviewer’s one.

Ultimately it comes down to this. If the authors would like to argue that simulated Holocene GMST trends were driven by CO₂ increases, that’s fine, I agree, but it’s well established and not novel. In the manuscript, the authors claim to detect a CO₂ warming mode in the simulation using EOF analysis, but the spatial pattern of that mode is both inconsistent with CO₂ increases (which should resemble supp Fig 4a), and very consistent with insolation-driven changes in temperature patterns (like supp Fig 4b), which don’t cause a net change in GMST, but are large changes and are a leading mode of variability. Coming back to their four reasons in support of their hypothesis, I’ll respond to each in turn:

1. (it is) in agreement with all other studies we are aware of That CO₂ drives the leading mode of variability (not the cause of the net warming) is not established (indeed, it’s the proposed novelty of this paper).
2. indicated by the high correlation between the warming mode (PC1) and the greenhouse effect The correlation with insolation is just as good.
3. because the annual global mean orbital forcing changes are marginal and about one to two orders of magnitude smaller than the changes in the greenhouse effect The large and asymmetric changes in seasonal insolation can drive large changes in the spatial pattern of temperature, making it the leading mode of variability, while averaging out to small annual changes (see supp fig 4b).
4. supported by additional sensitivity experiments showing that the warming

of the simulated global annual mean temperature induced by a 20 ppm CO₂ increase is about 0.3 K. In contrast, there is no substantial global annual mean temperature increase simulated due to changes in the orbital forcing

These sensitivity experiments clearly show that the spatial pattern associated with GHG increase does not match PC1; whereas the spatial pattern associated with insolation change matches clearly.

Again this seems to be a misunderstanding. All our arguments 1 to 4 were just given for the temporal evolution (PC1) and not for the spatial pattern (EOF1). In an earlier response (January 8, 2020) we already discussed the role of the orbital forcing for the spatial pattern of the warming mode (EOF1). Therefore, we discussed in the last response (March 17, 2020) only the time evolution of the warming mode (PC1). We wanted to make clear that the time evolution of the warming mode (PC1) is dominated by the CO₂ forcing. Our impression is that when we are talking about the temporal development/variability (PC1) this interpretation is also transferred to the spatial variability pattern (EOF1).

(a)

Prescribed idealised temperature field due to CO₂-forcing

Temperature evolution of a grid point

Prescribed idealised temperature field due to orbital forcing

extratropics

tropics

Temperature evolution of a grid point

Temperature increases in the tropics and decreases in the extratropics

EOF-analysis of the two added temperature fields

EOF1

PC1

Figure 1: The two upper panels show the prescribed temperature fields due to CO₂ and orbital forcing. The lower panel shows the EOF-analysis of the two added temperature fields. For more details see text.

Reviewers' comments, sixth round:

Reviewer #5 (Remarks to the Author):

Thanks to the authors for these revisions. I am comfortable with the manuscript following the revisions.